# Inter-comparison of tropospheric ozone column datasets from combined nadir and limb satellite observations

Carlo Arosio[1], Viktoria Sofieva[2], Andrea Orfanoz-Cheuquelaf[1], Alexei Rozanov[1], Klaus-Peter Heue[3], Diego Loyola[3], Edward Malina[4], Ryan M. Stauffer[5], David Tarasick[6], Roeland Van Malderen[7], Jerry R. Ziemke[5], and Mark Weber[1]

[1]Institute of Environmental Physics, Bremen University, Bremen, Germany
[2]Finnish Meteorological Institute, Helsinki, Finland
[3]Institut für Methodik der Fernerkundung am Deutschen Zentrum für Luft- und Raumfahrt (DLR)
[4]European Space Agency, ESRIN, Frascati, Italy
[5]NASA Goddard Space Flight Center, Greenbelt, MD, USA
[6]Environment and Climate Change Canada, Downsview, ON, CA
[7]Royal Meteorological Institute of Belgium, Brussels, Belgium

**Correspondence:** Carlo Arosio (carloarosio@iup.physik.uni-bremen.de)

**Abstract.** This manuscript presents an inter-comparison between existing tropospheric ozone column (TrOC) datasets obtained using combined limb and nadir observations, i.e. exploiting collocated stratospheric profile and total column information retrieved from limb and nadir satellite observations, respectively. In particular, seven datasets have been considered, covering the past two decades and consisting of monthly averaged time series with nearly global coverage. We perform a comparison in terms of climatology and seasonality, investigate the tropopause height used for the construction of each dataset and the related biases, and finally discuss long-term TrOC drifts and trends. The overall goal of the study is to assess the consistency between the datasets and explore possible strategies to reconcile the differences between them. Despite uncertainties associated with the limb-nadir residual methodology and large biases between the mean values of the considered datasets, we identify an overall agreement of TrOC distribution patterns. The different tropopause height definitions used to construct the datasets did not show a relevant role in explaining the biases between them. We demonstrate that a thorough investigation of the drifts with respect to ground-based observations is needed to evaluate TrOC trends from satellite data and that long-term trends in specific regions can be consistently detected, e.g. a positive trend of up to 1.5 DU per decade over China for the 2005-2021 period.

## 1 Introduction

Tropospheric ozone plays a crucial role in air quality and climate regulation, being a pollutant and a greenhouse gas (Warneck, 1999). It is responsible for respiratory problems, as it reduces lung function, and it has negative effects on ecosystems, e.g. inhibiting plant growth and reducing agricultural yields. Monitoring the concentration of tropospheric ozone is essential for understanding its impacts on health and climate (Mills et al., 2018; Fleming et al., 2018). Tropospheric ozone is a secondary pollutant produced by chemical reactions between nitrogen oxides ($NO_x$) and volatile organic compounds ($VOCs$) in the presence of sunlight ($\lambda < 420$ nm). The primary sources of tropospheric ozone precursors are both natural, e.g. wet-

land methane emissions, wildfires and lightning, and anthropogenic, e.g. vehicle emissions, industrial activities, and chemical solvents, making ozone a critical focus for air quality regulations and environmental health initiatives (Brown et al., 2013).

The direct retrieval of tropospheric ozone from nadir satellite observations has been investigated, starting with Global Ozone Monitoring Experiment (GOME) observations (Munro et al., 1998). More recently, tropospheric ozone retrievals have been performed, for example, using nadir measurements from GOME-2 (Miles et al., 2015), the Ozone Monitoring Instrument (OMI) (Bak et al., 2013), the Infrared Atmospheric Sounding Interferometer (IASI) (Boynard et al., 2009) and the TROPOspheric Monitoring Instrument (TROPOMI) (Mettig et al., 2021; Keppens et al., 2024). However, distinguishing between stratospheric and tropospheric ozone is challenging due to the low amount of ozone in the troposphere in comparison to the stratosphere, and the limited information content of nadir observations for vertical profiles.

Several residual methods involving the subtraction of the stratospheric ozone column (SOC) from the total column ozone (TCO) have been developed over the last decades. Cloud slicing is a technique that uses total column measurements taken at scenes with different cloud altitudes to derive vertical ozone profiles (Ziemke et al., 2001). The convective cloud differential (CCD) method is another technique used to measure tropospheric ozone (Ziemke et al., 1998). It exploits the presence of high convective clouds in the Pacific sector and assumes zonal invariance in stratospheric ozone. This method is particularly effective in tropical regions, where deep convective activity is frequent.

The limb-nadir tropospheric ozone residual (TOR) technique involves data from both limb-viewing and nadir-viewing satellite instruments, and provides tropospheric ozone column (TrOC) information. Limb-viewing instruments capture high-resolution vertical profiles by observing the atmosphere's limb, while nadir-viewing geometry offers higher horizontal sampling. This approach was first proposed by Fishman and Larsen (1987) for a combination of observations from the Total Ozone Mapping Spectrometer (TOMS) and the Stratospheric Aerosol and Gas Experiment (SAGE). In the last three decades this has been applied to several instrument combinations and this paper focuses on datasets retrieved using combinations of limb and nadir satellite observations (or reanalysis data) covering the last two decades.

Satellite observations over the past few decades have shown significant trends in tropospheric ozone levels (Gaudel et al., 2018). Regional variations are driven by factors such as anthropogenic emissions, including nitrogen oxides ($NO_x$) from industrial activity, and natural sources like wildfires and biogenic emissions. In many regions, especially in developing countries, tropospheric ozone levels have been increasing due to rising pollution levels, while in parts of North America and Europe ground ozone levels have stabilized or decreased thanks to air quality regulations (Pope et al., 2023, 2024). Understanding these trends is relevant for developing strategies to mitigate ozone pollution and to evaluate the success of such mitigation strategies, which may be costly. Despite the number of techniques, measurements and research conducted, reconciling the differences between satellite and in situ observations has been challenging (Tarasick et al., 2019).

This work fits within the framework of the Tropospheric Ozone Assessment Report (TOAR) II initiative, which has the aim of providing an up-to-date scientific assessment of the global distribution of tropospheric ozone and its trends from ground-based instruments and satellite observations. An overview of tropospheric ozone trends was given by Gaudel et al. (2018), concluding that satellite data were suitable for computing trends, but discrepancies between different datasets were large, with the need for further studies to reconcile those differences. Lately, a few studies tried to reconcile these discrepancies, such as

Gaudel et al. (2024) for the tropics, pointing out the need to maintain and develop high-frequency continuous observations, and Pope et al. (2024) that show small linear trends at mid-latitudes affected by large inter-annual variability.

The manuscript is structured as follows: Sect. 2 describes the datasets used for this analysis and gives more insights into the limb-nadir TOR technique. Sect. 3 presents a comparison of the selected TOR datasets in terms of climatology, anomalies and seasonality. The role of the tropopause height definition used to construct the datasets is discussed in Sect. 4, where a method

to correct the tropopause definition-related bias between time series is presented and assessed. In Sect. 5, a comparison with ozonesondes is presented; these independent measurements are also used to evaluate possible drifts in the datasets. Finally, in Sect. 6 the focus is on long-term trend studies, by using the datasets covering a period of at least 10 years.

## 2   Limb-nadir combined datasets

For this inter-comparison study we consider seven limb-nadir TOR datasets, as listed in Table 1, where the time frame of each

product, the chosen tropopause height (TPH) definition and the horizontal resolution are listed. The choice of the TPH plays a relevant role for the construction of these datasets, as it is used as a bottom boundary for the integration of the stratospheric profile. If a gap between the two is present, a climatology or a model needs to be used to extend the profiles down to the TPH. Discrepancies in TPH are not linear in ozone due to its increasing concentration with altitude in the stratosphere; in addition, the chosen TPH definition impacts the sensitivity of the TrOC product to stratospheric ozone, as the ozone profile generally

starts to increase below the typical thermal tropopause (e.g. Monsees et al., 2024). In this study, satellite datasets are monthly Level 3 (L3, gridded) time series.

**Table 1.** Limb-nadir TOR datasets included in this study.

| Name | Time frame | TPH definition | Original horizontal resolution |
|---|---|---|---|
| OMI-LIMB (Sofieva et al., 2022) | 2004-2023 | Thermal (WMO) or ozonopause | 1°x1° |
| GTO-LIMB (Sofieva et al., 2022) | 2004-2023 | Thermal or ozonopause | 1°x1° |
| S5P-BASCOE (Heue et al., 2022) | 2018-2023 | 380 K (tropics), 3.5 PVU (extra-tropics) | 0.25°x0.25° |
| SCIA+OMPS (Orfanoz-Cheuquelaf et al., 2023) | 2004-2023 | Thermal (tropics), 3.5 PVU (extra-tropics) | 5°x5° |
| OMI-MLS (Ziemke et al., 2006) | 2004-2023 | Thermal | 5°x5° |
| OMPS-MERRA (Ziemke et al., 2022) | 2012-2023 | 2.5 PVU or 380 K (whichever lower) | 1°x1° |
| EPIC-MERRA (Ziemke et al., 2022) | 2015-2023 | 2.5 PVU or 380 K (whichever lower) | 1°x1° |

In the following the various datasets are briefly described.

OMI-LIMB and GTO-LIMB are two datasets developed at the Finnish Meteorological Institute (FMI) as part of the SUNLIT project (Sofieva et al., 2022) and within the ESA Climate Change Initiative (CCI). These datasets combine total ozone columns

(TOC) either from NASA's OMI or from GOME-type Ozone (GTO) data (Coldewey-Egbers et al., 2022) with limb information coming from several instruments, such as MLS (Microwave Limb Sounder), OSIRIS (Optical Spectrograph and InfraRed

Imaging System), MIPAS (Michelson Interferometer for Passive Atmospheric Sounding), SCIAMACHY (SCanning Imaging Spectrometer for Atmospheric CHartographY), OMPS-LP (Ozone Mapping and Profiles Suite – Limb Profiler), and GOMOS (Global Ozone Monitoring by Occultation of Stars). The ozone profiles were merged into a high-resolution dataset (LIMB-HIRES), which was used for computation of the stratospheric ozone columns (Sofieva et al., 2022). The MLS record is used as a reference in creating the LIMB-HIRES dataset. Two tropospheric ozone columns are provided in both OMI-LIMB and GTO-LIMB datasets: from ground to the thermal tropopause and from ground to 3 km below the thermal tropopause.

The S5P-BASCOE dataset (Heue et al., 2022) is generated through the combination of TROPOMI/Sentinel-5 Precursor (S5P) TOC (Garane et al., 2019) and the Belgian Assimilation System for Chemical ObsErvations (BASCOE), which assimilates MLS and other stratospheric ozone profiles into a chemical transport model to separate the stratospheric and tropospheric ozone components. The time series has a high spatial resolution and covers the period from 2018 to the present. An isentropic tropopause definition was used in the tropics, whereas in the extratropics a dynamical tropopause was used.

The SCIA+OMPS dataset is the only merged product of the list, as the TrOC datasets independently retrieved for SCIAMACHY and OMPS measurements have been merged on a monthly gridded basis. The retrieval of OMPS (Orfanoz-Cheuquelaf et al., 2023) and SCIAMACHY (Ebojie et al., 2016) TrOC data employs a specific TOR methodology called limb-nadir matching (LNM), which consists in combining two observations of nearly the same air mass performed by the same instrument (SCIAMACHY) or from two instruments on the same platform (OMPS). This technique has the advantage of minimizing instrument-related biases. The merging of the recently re-processed SCIAMACHY with LNM OMPS TrOC was performed on de-seasonalized anomalies of the two time series by using OMI-MLS as a transfer function: the bias between SCIAMACHY and OMI-MLS anomalies in the period 2007-2012 and between OMPS and OMI-MLS over 2012-2017 was removed to merge them. OMPS TrOC seasonal cycle was then added back to get a dataset in DU covering 2004-2023.

The OMI-MLS TrOC dataset (Ziemke et al., 2006) provides global measurements of tropospheric ozone by combining data from OMI and MLS, both aboard the Aura satellite. The dataset provides monthly L3 data from $60°$S to $60°$N at a resolution of $5°$ latitude by $5°$ longitude. To take into account an identified drift in OMI time series, this dataset was corrected by the data provider by adding a drift at a post-processing step, as described in Gaudel et al. (2024). This data spans from October 2004 to the present and was used in several studies to investigate long-term trends, pollution events, and interactions with other atmospheric gases (Ziemke et al., 2019, 2022). The dataset adopts the WMO thermal TPH definition (WMO, 1957), i.e. based on the $2\,\mathrm{K\,km^{-1}}$ thermal vertical gradient thershold, using NCEP reanalyses.

For the two OMPS-MERRA and EPIC-MERRA datasets, SOC information is derived by vertically integrating Global Modeling and Assimilation Office (GMAO) Modern-Era Retrospective analysis for Research and Applications-2 (MERRA-2) assimilated Aura MLS ozone profiles (Gelaro et al., 2017; Wargan et al., 2017), from the top of the atmosphere down to the tropopause. This is combined with TCO from OMPS and the Deep Space Climate ObserVatoRy (DSCOVR) EPIC (Marshak et al., 2018) instruments, respectively. The dynamical TPH definition is used in both cases, with a 2.5 PVU or 380 K threshold, from MERRA-2 data.

All datasets are monthly averaged TrOC values and have been binned to the same spatial resolution, $5°$ latitude x $5°$ longitude, for the following analysis.

## 3 Comparison of the datasets and their climatologies

A first comparison between the datasets is performed in terms of zonal averages to get a rough assessment of their biases, as shown in Fig. 1 for several latitude bands defined according to the TOAR II Guidelines. The figure showcases the presence of large discrepancies between the datasets, also in terms of seasonal cycle. For example, we notice the generally lower TrOC values from GTO-LIMB and OMI-LIMB, including their more pronounced seasonal cycle at mid-latitudes with respect to the other datasets. Evident is also the different seasonal cycle shown by OMI-MLS at southern mid-latitudes or the generally high bias of S5P-BASCOE TrOC above 40° latitude. The best absolute agreement between all datasets is found during summer at northern mid-latitudes. Large discrepancies are visible at southern mid-latitudes, especially in the last three years of the time series.

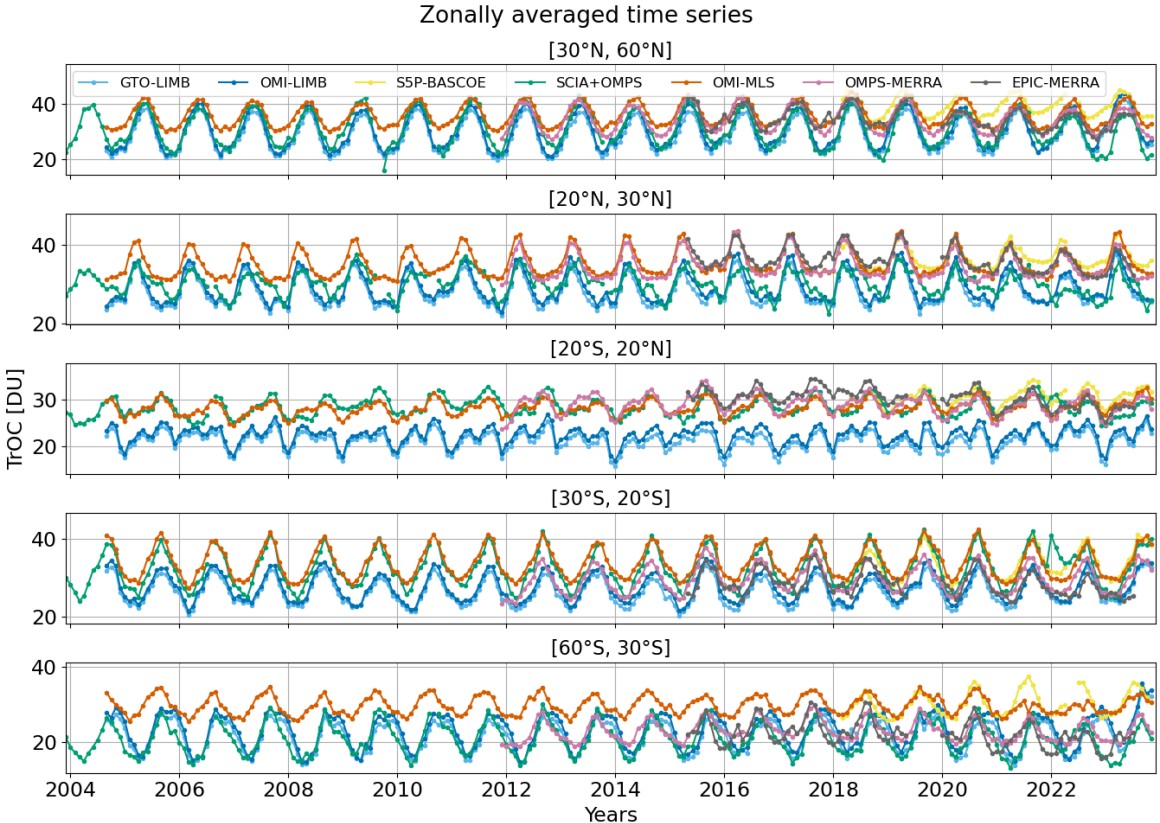

**Figure 1.** Time series of zonal mean TrOC averaged in several latitude bands.

These biases have several possible reasons: e.g., overall discrepancies in SOC and TOC between satellite products, the specific tropopause definition adopted to construct the TOR product, the climatology or model used to fill SOC gaps, and the

criteria used for the combination of SOC and TOC. For the rest of the paper, we aim to change the perspective to highlight similarities and possibly resolve the differences.

We then performed a comparison in terms of climatologies. In the Supplements, Fig. S1 provides a comparison of the annual mean climatologies. Here Fig. 2 displays mean climatologies of the datasets for each season. Values are averaged over each respective time period, so that discrepancies may also arise for this reason. To better assess the common ozone patterns, the climatologies have been de-biased with respect to the multi-instrumental mean so that the global mean value [60°S, 60°N] for all maps in each column is the same (and reported on top). We focus here on common features of TrOC that are evident for all datasets, in particular, the wave-1 pattern in the tropics, as visible in Fig. 2. This is a clear zonal asymmetry in the TrOC distribution with higher ozone concentrations over the Atlantic and African regions, and lower concentrations over the Pacific and Indian Oceans (Fishman et al., 1990). The main reasons for this pattern are related to the large biomass burning taking place in the African and South American regions, but also the weaker intensity of deep convection in the Atlantic sector. This pattern is influenced by El Niño–Southern Oscillation (ENSO) and by the intensity of the Intertropical Convergence Zone (ITCZ) (Bruckner et al., 2024). Regional hot-spots of TrOC are visible over polluted areas, where precursor emissions are high, such as over China, Southern Europe, between the Arabian sea and India, and the west coast of US. Low ozone concentrations are typical over the oceans and unpolluted regions due to limited precursor availability. Comparing the rows of the figure, we notice datasets displaying a large ozone seasonality, e.g. OMI-LIMB and GTO-LIMB that show the largest summer TrOC values at northern mid-latitudes. S5P-BASCOE shows higher ozone values in winter at northern high latitudes, whereas EPIC-MERRA has unusually low ones at southern high latitudes. In the tropics, SCIA+OMPS and OMPS-MERRA display less pronounced seasonality and longitudinal structure with respect, for example, to OMI-MLS or OMI-LIMB.

The seasonality is further investigated in Fig. 3 by plotting the seasonal cycle of the datasets averaged in several latitude bands. The mean seasonal cycle is defined as follows:

$$SC_m = \frac{1}{N_m} \sum_{t=1}^{N_m} TrOC(t), \tag{1}$$

where $N_m$ is the number of available monthly mean values $TrOC(t)$ for each specific month of the year $m$, e.g. January, in each time series. The offsets between the datasets have been removed, bringing all seasonal cycles to the same yearly average value, corresponding to the satellite-ensemble average, for a better comparison.

TrOC values are generally higher in the Northern Hemisphere during summer due to increased photochemical production and lower in winter. In the Southern Hemisphere, seasonal patterns are less pronounced. We notice that at most latitude bins the zonally averaged seasonality is in very good agreement between the datasets. The worst agreement is visible at southern mid-latitudes, with OMI-LIMB and GTO-LIMB showing a stronger seasonal cycle with respect to the others. In addition, a large scatter is visible in austral winter-time (JJA).

Another comparison related to the temporal evolution of the data products is shown in Fig. 4, which displays the time series of the absolute anomalies as a function of latitude, which are defined as follows:

$$\Delta(t_m) = TrOC(t_m) - SC_m, \tag{2}$$

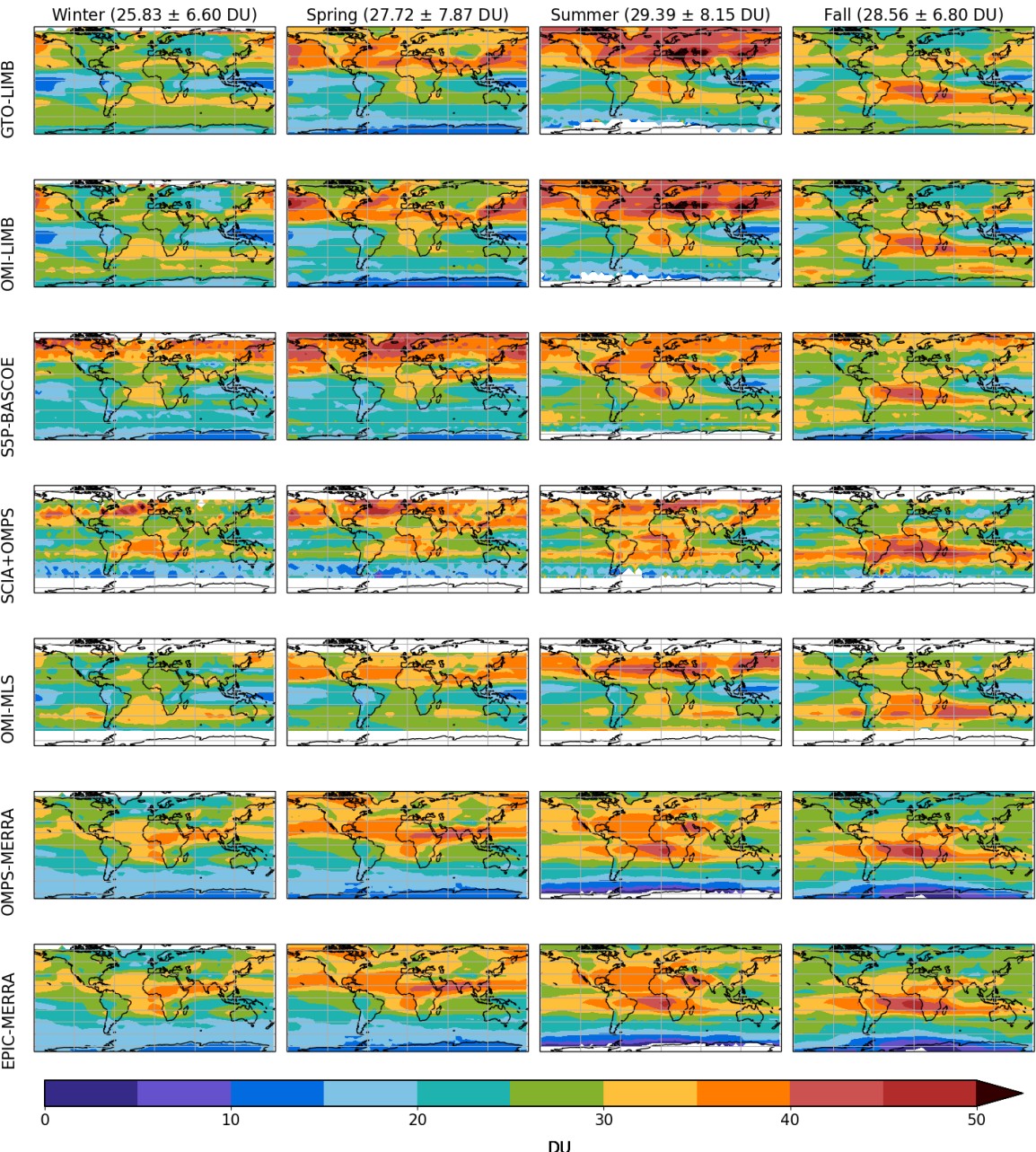

**Figure 2.** Climatology of the considered datasets, averaged over each season and after debiasing them to the same mean value, i.e. after bringing them to the multi-instrument global mean. Titles include multi-instrument mean TrOC values and corresponding standard deviations for each season.

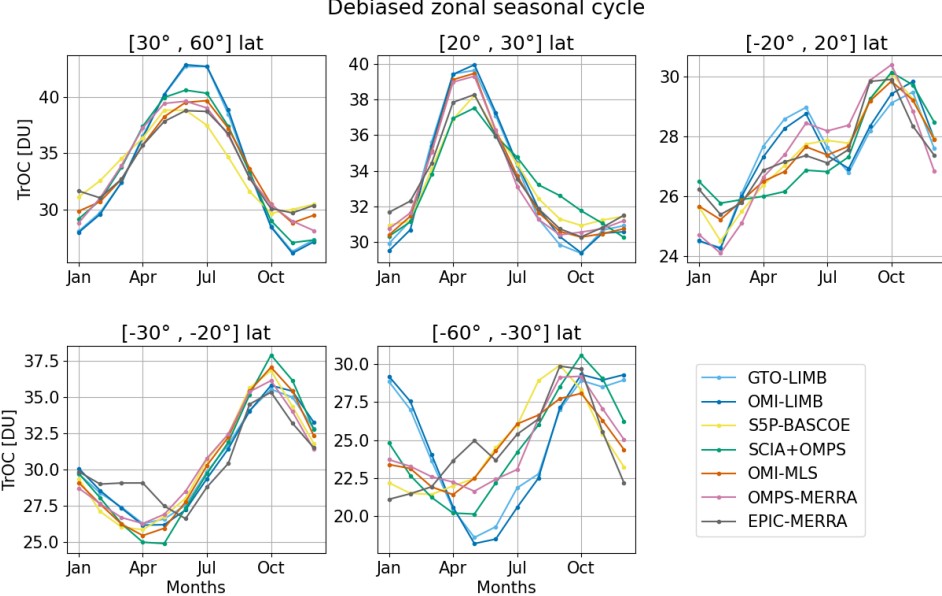

**Figure 3.** Seasonal cycle of the seven considered datasets (over their respective time coverage) in several latitude bands, after de-biasing to the multi-instrument mean for each panel.

where $m$ indicates the month of the year, e.g. January, and $t_m$ all months, e.g. all Januaries, in the time series.

From Fig. 4, we can assess the presence of outliers in the data, as well as discontinuities or anomalous periods. In this respect, a pronounced feature is related to the drop that occurred in 2020, with negative anomalies up to 3-4 DU visible, especially in OMPS-MERRA and EPIC-MERRA, but also present in the other datasets with a smaller magnitude (1-2 DU). Ziemke et al. (2022) described a drop in ozone values at northern mid-latitudes starting from 2020 and attributed it to the COVID-19 pandemic. Ground-based observations have detected a small drop at some stations of up to 1-2 DU (Steinbrecht et al., 2021), with hints of a rebound over North America after 2021 (Chang et al., 2023a). Other studies recently addressed this topic, providing the chemical background to understand this drop. Miyazaki et al. (2021) estimated an ozone loss of up to 5 ppb in the lock-down months in 2020 due to the reduction in $NO_x$ anthropogenic emission. Putero et al. (2023) found negative ozone anomalies at high-elevation sites in North America and western Europe, particularly in 2020, confirmed also by Chang et al. (2022), Elshorbany et al. (2024) and Pimlott et al. (2024).

In our datasets, we found that when changing the period for calculating the seasonal cycle, this drop becomes evident in other datasets as well: Fig. S2 shows anomalies as in Fig. 4, but with the seasonal cycle computed over 2016-2019, instead of using the entire period of the respective time series. Fig. S3 better showcases the negative anomalies (example time series at 42.5°N) in recent winters also for GTO-LIMB and OMI-LIMB when the 2016-2019 is subtracted. The presence of a such large drop in some of the considered datasets is still under investigation. We looked for the presence of any sudden change in reanalysis temperature and stratospheric ozone without finding convincing evidences of artefacts that could explain this pattern. Also in

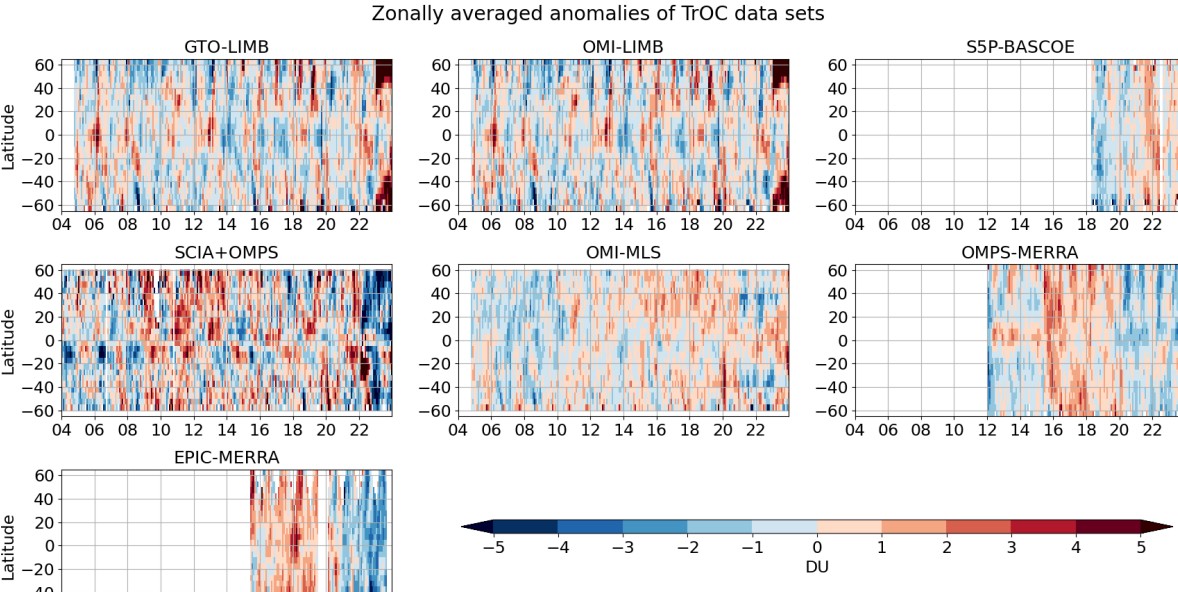

**Figure 4.** De-seasonalized time series (with respect to each respective period) as a function of latitude for the seven considered datasets.

TPH anomalies no evident discontinuities have been found around 2020 (as shown Fig. S4), without indications of TPH-related artifacts. Further studies are needed to better understand this feature and the discrepancies between datasets.

Apart from this drop, a high-ozone artefact related to the Hunga-Tonga volcanic eruption is visible in Fig. 4, particularly in SCIA+OMPS data, most probably coming from a sub-optimal filtering of the stratospheric column data. This dataset also shows a noisier time series than the others. A Quasi Biennial Oscillation signature was detected particularly in GTO-LIMB and OMI-LIMB datasets, which is most probably a SOC interference. This signature is not well visible from this figure but is shortly discussed in Sect. 6. A positive TrOC trend can be identified by eye in the OMI-MLS dataset, but this is not evident for

other datasets. Unexpectedly large anomalies in SCIA+OMPS, GTO-LIMB and OMI-LIMB in 2023 are under investigation.

## 4 Role of the TPH discrepancies and possible corrections

The definition of TPH used in the construction of the datasets, listed in Table 1, may play an important role in the biases between them. As already mentioned, discrepancies in TPH influence TrOC due to the strong ozone gradient at this altitude. Figure 5 shows the zonal averaged values of the TPH (in hPa) for the selected datasets, each over the respective time frame.

Differences are largest in the sub-tropical and mid-latitude regions, particularly for OMPS-MERRA (and EPIC-MERRA, using the same TPH data), which adopts a dynamical definition of 3.5 PVU.

For some datasets, such as SCIA+OMPS, another impacting factor is related to the climatology used to complete the stratospheric limb profiles in case the lowest measurement point lies above the TPH. In this way, depending on the limb vertical range and the used TPH definition, the adopted climatology will play a more or less relevant role.

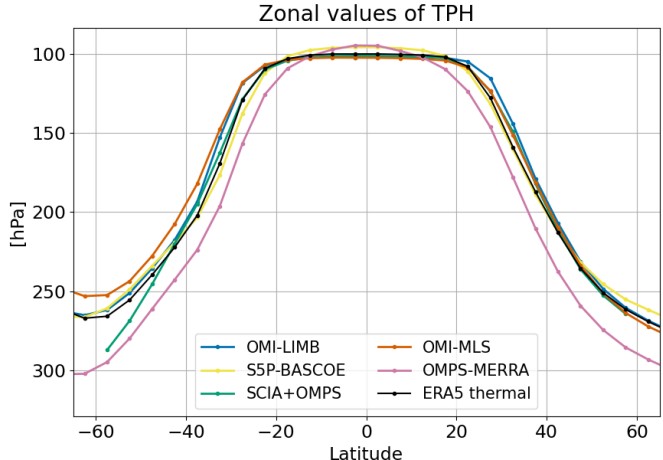

**Figure 5.** Zonal mean TPH (pressure, hPa) for the considered datasets, including ERA5 thermal tropopause values.

It is clear from comparing this figure with Fig. 1 that the main biases between TrOC are not related to the tropopause definition. For example, OMPS-MERRA and EPIC-MERRA have the lowest TPH at northern mid-latitudes, whereas the tropospheric ozone column is the largest, opposite to expectation. Nevertheless, we investigated an approach to remove the biases between the datasets related to the different TPH definitions by subtracting the ozone sub-columns corresponding to the gap in TPH. This was done using the European Centre for Medium-Range Weather Forecasts (ECMWF) ERA5 monthly

gridded ozone profiles at each latitude and longitude bin. As reference TPH, the ERA5 dataset was selected. The chosen time period for this analysis is 2018-2022 as it is covered by all datasets. Figure 6 shows in panels (a) and (c) an example of TrOC time series for the selected datasets over the chosen period, zonally averaged at 45°N, respectively before and after subtracting the column gaps displayed in panel (b). We subtract the column gaps averaged over the selected period, for each latitude/longitude bin. As one can see in the bottom panels, the standard deviation of the mean values of the datasets in each

latitude-longitude bin generally increases after applying the correction. This is caused, as seen comparing panels (a) and (c), by

the larger correction needed for the OMPS-MERRA and EPIC-MERRA lines, bringing them to a lower level with respect to the other time series. An exception is the northern subtropical band, where the standard deviation after the correction decreases, indicating that, at least in this region, the biases between datasets are mostly caused by TPH discrepancies.

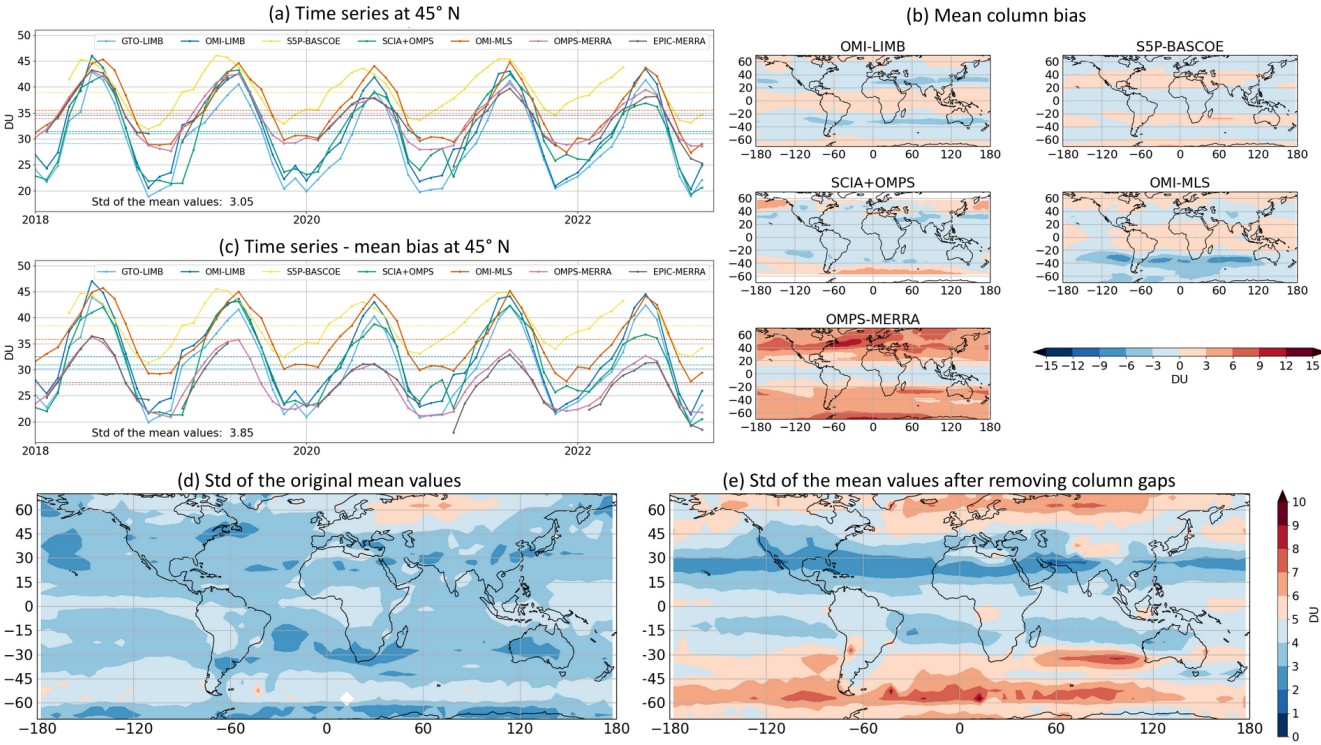

**Figure 6.** Panels (a) and (c) show an example of TrOC time series from 2018 to 2022, zonally averaged at 45°N, respectively before and after subtracting the column gaps displayed in panel (b). The bottom panels (d) and (e) show the standard deviation of the mean of the datasets in each bin, respectively before and after applying the correction.

These results show that this methodology does not consistently reduce the bias between the time series. We also tested the removal of monthly-resolved TPH-related biases, i.e. removing seasonal cycle biases, without improving the overall agreement of the datasets either. It would be useful to test this correction on L2 data, which is however out of the scope of this study, as we analyze L3 data only. However, in Fig. S5, we show a hint of the better agreement that we could obtain with a L2 correction using SCIAMACHY TrOC data.

Another important aspect of the TPH having an impact on the TOR product is related to their possible long-term drift. To assess this aspect, we computed linear trends of the deseasonalized anomalies of the TPH time series, zonally averaged over the period 2005-2021. Figure 7 displays these linear trends for the products with the longer time series, as reported in the legend, including the ERA5 thermal tropopause data. Shaded areas are the 2-$\sigma$ uncertainties. GTO-LIMB is not included as the results are very similar to those of OMI-LIMB. Trends are generally within $\pm 3$ hPa per decade, with a fairly common pattern as a

function of latitude for the different datasets, except for the southern mid-latitudes, where a larger scatter is found and even
the sign of the trend changes between the datasets. These differences are possibly related to different sampling of the satellite
observations at these latitudes, to the different TPH definitions, and possibly to a discrepancy between ERA5 and MERRA-2
reanalysis. In addition, SCIA+OMPS is showing some larger deviations from the others due to sampling issues.

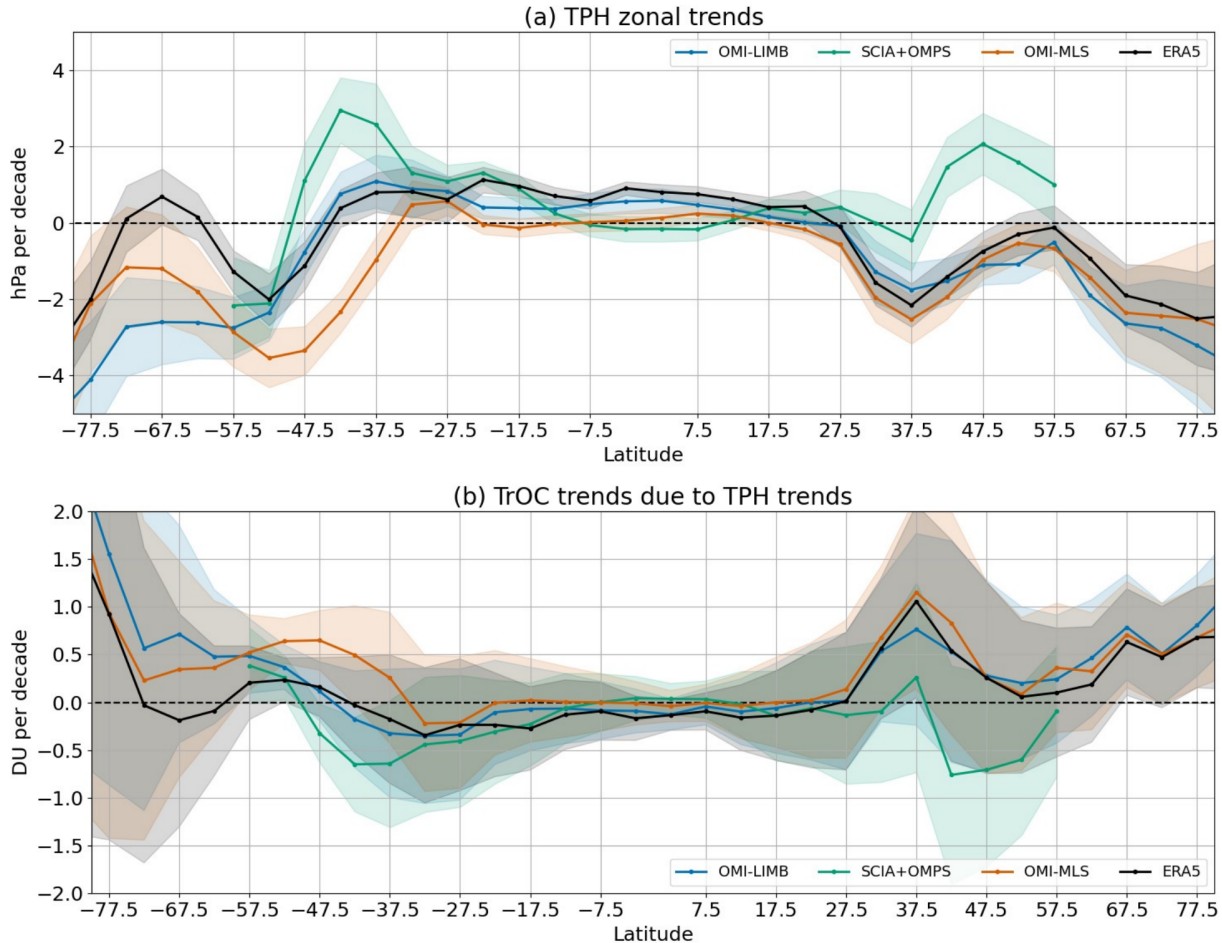

**Figure 7.** Linear trends in hPa per decade of the TPH from the datasets, including ERA5 monthly time series, as a function of latitude. In the
bottom panel the respective TrOC trends are shown in DU per decade.

In the bottom panel of Fig. 7, the TrOC trends due to TPH trends only (from panel a) are determined using ozone data from
ERA5, following these three steps:

1. The ERA5 TrOC time series was derived by integrating the ozone profiles up to the corresponding thermal TPH (from
   ERA5 temperature profiles).

2. TPH trends from the individual datasets (from panel a) were added to the ERA5 TPH time series and the ERA5 TrOC time series was re-calculated by using the adjusted TPHs values.

3. The linear fit to the difference between both ERA5 TrOC time series was calculated and is shown in panel (b).

The TPH-related TrOC trends are generally close to zero in the tropics, within $\pm 0.5$ DU per decade in the [-30°S, 30°N] band. On the contrary, in the sub-tropics and at mid-latitudes both values and the discrepancy between the datasets get larger, with a peak around 37.5°N of about +1 DU per decade. These results should be taken into consideration when assessing TrOC trends from the datasets.

## 5 Comparison with HEGIFTOM sondes and relative drift

The HEGIFTOM (Harmonization and Evaluation of Ground-based Instruments for Free Tropospheric Ozone Measurements) working group aims at evaluating and harmonizing tropospheric ozone data obtained from different observing networks of ground-based instruments in order to reconcile the differences in ozone distribution and trends between the different ground-based platforms (Van Malderen and Smit, 2020). For the present analysis, we are using monthly mean values of TrOC provided for the sonde stations listed in Table S1 in the Supplements. The TrOC was derived by integrating the sonde profiles from 235 ground to the thermal tropopause level, which was derived by the data provider from the ozonesonde temperature profiles using the WMO definition.

For the present study, the main aim of using the HEGIFTOM TrOC time series is to provide an assessment of potential drifts affecting the satellite TrOC datasets. The drift is defined as the linear trend of the difference between the satellite data and the reference HEGIFTOM time series.

For the drift assessment, we followed the procedure described here: for each available HEGITFTOM sonde station (see Table S1), we found the corresponding satellite data grid cell containing the location of the station and computed de-seasonalized (absolute) anomalies for both time series. For each station, we then computed differences between anomalies (TOR - HEGIFTOM). The linear trend of these differences corresponds, as already mentioned, to the drift of the satellite product with respect to sonde observations. To minimize the noise, we focused on two latitude bands, i.e. tropics [-30°S, 30°N] and northern mid-latitudes 245 [40°N, 60°N], where enough sonde stations are available for taking the mean. We discarded sonde stations with a particularly short or sparse record and satellite datasets having a short time span, i.e. S5P-BASCOE and EPIC-MERRA. For the remaining TrOC products, Fig. 8 shows the time series of the differences and their respective linear trends, i.e. the drifts of the satellite datasets, averaged over the two selected latitude bands. Both panels display the monthly time series and their 13-month running averages for a less noisy visualization of the long-term tendency.

At northern mid-latitudes (panel a) the drifts for three datasets, i.e. OMI-LIMB, GTO-LIMB and OMI-MLS are fairly close to zero, with p-values > 0.33, which corresponds to very low probability of a drift. SCIA+OMPS is affected by larger oscillations in the difference to HEGIFTOM and shows a negative drift, still with a p-value > 0.1, i.e. low confidence of a drift. OMPS-MERRA has only an 11-year time span, over which it displays "highly certain" drift.

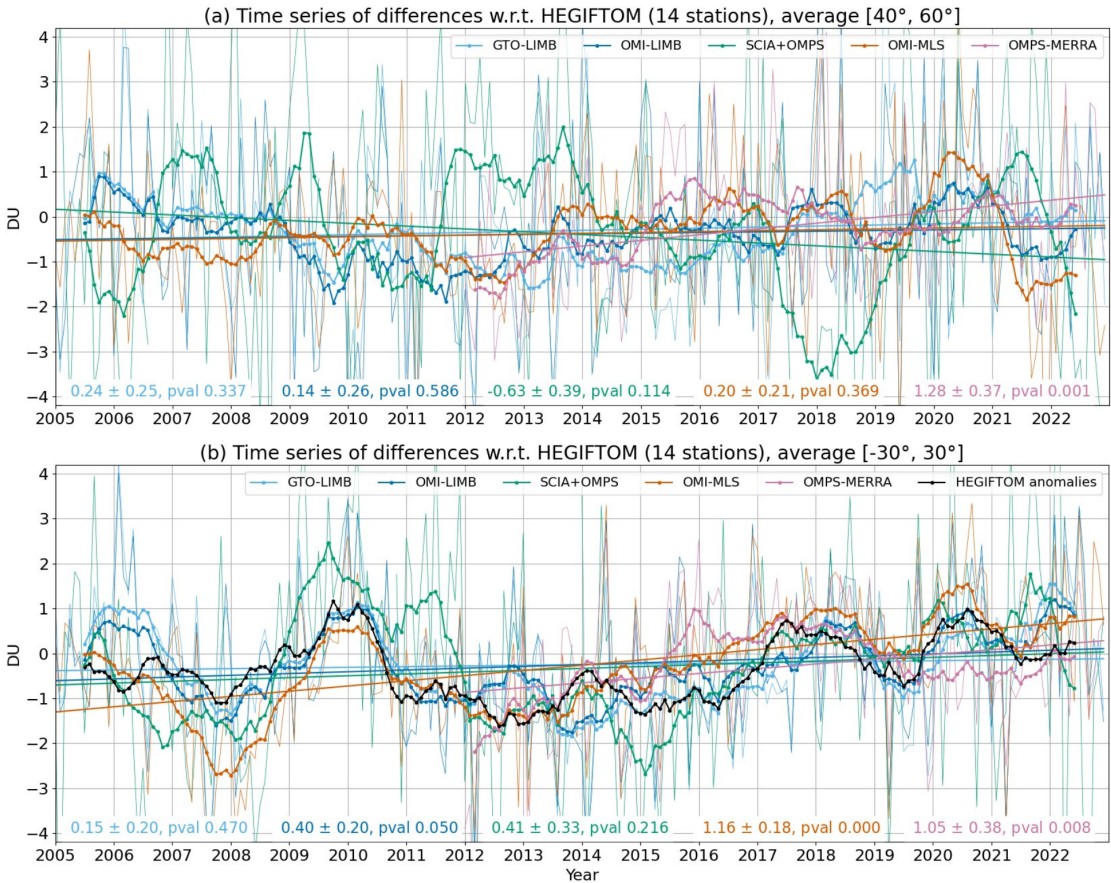

**Figure 8.** Drift in $DU/dec$ of the difference time series between satellite and HEGIFTOM sonde anomalies. Thin lines indicate the monthly difference time series, whereas thicker lines are their 13-month averages; in panel (b) the HEGIFTOM anomalies are also shown (with sign changed). Drift values with corresponding $2\sigma$ uncertainties and p-values are reported at the bottom of the panels. The titles include the number of stations used for the analysis.

In the tropics (panel b) we notice a common pattern in the differences among the five datasets, all affected by a positive drift
with pronounced oscillations. For OMI-MLS and OMPS-MERRA the drift has a p-value $< 0.01$ (i.e. "very high confidence" of a drift), especially over the last 10 years. OMI-LIMB shows the lowest drift with a p-value $> 0.33$, i.e. "very low confidence". Over-plotted in black is the HEGIFTOM anomaly time series, changed of sign: the similarity between this line and the satellite-to-sonde differences indicates that these patterns are not captured by the satellite datasets. We further investigated the presence of these patterns by applying a weighting to the ozonesondes with the aim to test the influence of the lowermost troposphere
on these patterns, as we know that the sensitivity of nadir observations decreases in the lowermost troposphere (Sofieva et al., 2022). This is discussed in Appendix A. Fig.A1 shows that the pattern described in the tropics is partially reduced by reducing the contribution from the lowermost troposphere. In this case, drifts are closer to zero (except for SCIA+OMPS) with p-values

larger than 0.15 for OMPS-MERRA, OMI-LIMB and GTO-LIMB. This suggests that for investigating the stability of satellite-based time series with respect to high vertical resolution observations, the application of averaging kernels could be beneficial, to take into account the different sensitivity with respect to satellite measurements.

An overall inter-comparison between HEGIFTOM and satellite data to provide a general assessment of the absolute bias and scatter between them and their trends is described in Appendix B.

## 6   Trends in geographical regions

Studies on tropospheric ozone trends from ozonesonde measurements, such as Christiansen et al. (2022) analyzing trends over the 1990–2017 period and Wang et al. (2022) over the 1995-2017 period found generally positive trends in the free troposphere and larger positive values in the lower troposphere in Southeast Asia, also confirmed by Stauffer et al. (2024) over the 1998–2022 period. Recently, Van Malderen et al. (2025) performed a thorough analysis of the HEGIFTOM dataset, assessing trends over the period 2000-2022 from ozonesonde stations globally distributed and comparing different trend calculation methods. The authors concluded that TrOC trends generally lie within the [-3 ppb/dec; +3 ppb/dec], with difficulties in finding common consistent geographical patterns.

Long-term satellite-detected ozone trends were discussed recently by several studies. Pope et al. (2023) investigated changes in the lower tropospheric column over the period 1996-2017, finding positive trends with high confidence in the tropics and with lower confidence at mid-latitudes. Froidevaux et al. (2025) showed that MLS-based upper tropospheric trends over the 2005-2020 period are consistent with TrOC trends in the tropics (Ziemke et al., 2019), with positive values over Southeast Asia and the sub-tropical Atlantic region.

We explored the changes in TrOC from three satellite data records, i.e. OMI-LIMB, SCIA+OMPS and OMI-MLS, which cover the longest time frame. The chosen period is 2005-2021 to avoid possible perturbations from the Hunga-Tonga eruption in 2022 and unexplained features in some datasets in 2023 (as pointed out in Sect. 3). GTO-LIMB has shown very similar trends to OMI-LIMB and has not been included in the next figures. Instead of considering latitudinal averages, we focused on specific geographical regions, which are of interest to human-related activities and their changes over the last decades. In order to define the extent of these regions, we investigated the spatial correlation of the seasonal cycle of the chosen datasets, requiring a high homogeneity of the seasonality within each region and between the datasets. An example is shown in Fig. S6.

Figure 9 displays the defined geographical regions of interest, which are:

- US, where stringent policies where introduced to reduce air pollution;

- the Mediterranean region, typically affected by high summer levels of ozone;

- China, where we expect an increase in TrOC values as reported by the sondes;

- the Atlantic Ocean off the African coast, a region affected by transport of precursors from wild fires and characterized by high levels of tropospheric ozone;

– the Amazon area, because of the change in land use and deforestation.

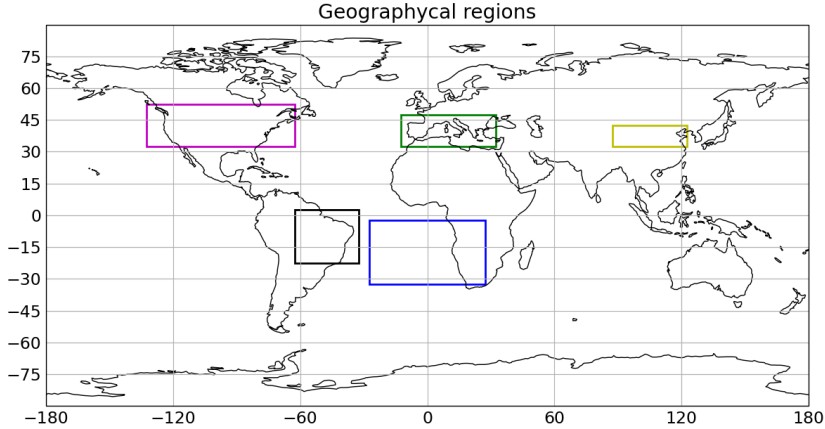

**Figure 9.** Geographical regions selected for the trend study.

From the time series averaged in each geographical region, we removed the seasonal cycle (Eq. 2, to get absolute anomalies) and we applied a quantile regression (QR) model, as recommended by TOAR-II to reduce the impact of outliers on the regressed trends. In the regression model, we included three proxies: the first two principal components of the QBO and the Multivariate El Nino Southern Oscillation (ENSO) Index (MEI), without any lag. The latter in particular has a relevant impact on ozone variations in the tropical regions (e.g. Rowlinson et al., 2019). We include the QBO to account for its influence on the SOC:

Fig. S7 shows the fit contributions of these two proxies in the tropics for OMI-LIMB data. We also tested the use of different proxies, such as the aerosol optical thickness and the TPH time series, without finding a significant contribution of these proxies to the trends.

     Figure 10 shows the absolute anomaly time series and the respective trends in the five defined regions for OMI-LIMB, SCIA+OMPS and OMI-MLS, over the period 2005-2021. The subplots show in thicker lines the 13-month running mean of

the dataset time series averaged within each defined region after subtracting the fit contributions from QBO and ENSO. The respective QR trend values and uncertainties are also reported. Table 2 summarizes the results per region and reports the trend values in $\mathrm{ppb}$ per $\mathrm{decade}$ as well [1], with p-values and confidence.

     The only region with a positive trend in all datasets is China with values up to +1.5 DU per decade for OMI-MLS (p-value < 0.01, i.e. very high confidence) and closer to zero for SCIA+OMPS (p-value > 0.33, i.e. very low confidence), with a hint

of a possible change in trend sign from 2017. This is consistent with Froidevaux et al. (2025); Gaudel et al. (2024) and in agreement with Lu et al. (2024) using ozonesondes and IAGOS (In-service Aircraft for a Global Observing System) data. However, we need to take into consideration the presence of a positive TPH-related trend, described in Fig. 7, which reduces the confidence of the found positive trends. Positive trends are also seen by SCIA+OMPS and OMPS-MLS in the Amazon but

---

[1]The conversion DU to ppb was done with the following approximations: $1\ DU = 2.14 * 10^4\ \mu g(O_3)/m^2$, assuming a 10 km layer and $1\ \mu g/m^3 = 0.5$ ppbv

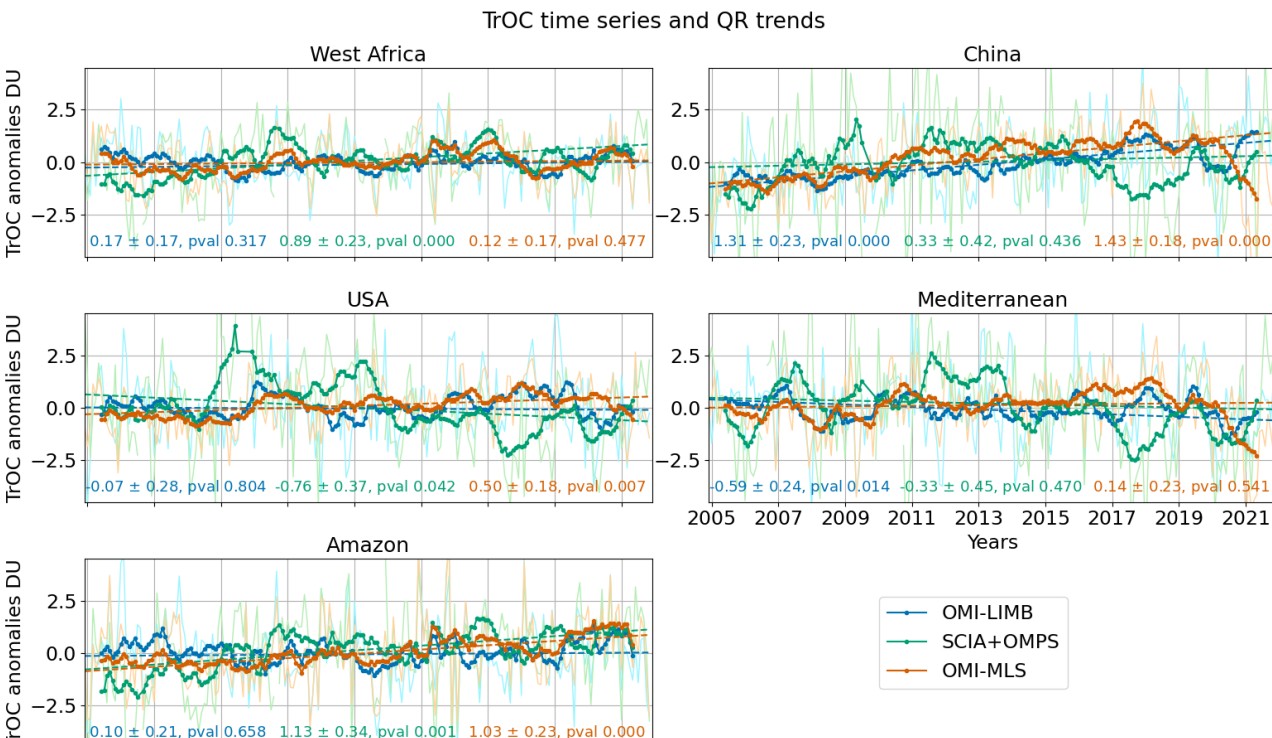

**Figure 10.** TrOC time series in terms of absolute anomalies (with QBO and ENSO fit contributions subtracted) averaged in specific regions for three datasets over the period 2005-2021. Linear fits are over-plotted and trends reported in DU per decade.

are not confirmed by OMI-LIMB. OMI-MLS also shows a positive drift with respect to sondes in the tropics, which gives less confidence to this value. Close to zero trends are found in the West Africa region and the US (except for SCIA+OMPS, due to large oscillations). Summer-time trends over the US are also close to zero (not shown). The Mediterranean region shows negative or close to zero trends for OMI-MLS, but with p-values > 0.2 (low confidence). These negative trends are possibly related to the EU policies introduced to improve air quality. We further investigated trends in Europe, as shown in Fig. S8, where negative low-confidence trends are observed. In particular, close to zero summer-time changes are detected over Europe, whereas, over the Mediterranean, all three datasets show negative summer-time trends, although with large p-values. Due to the possible positive TPH-related ozone trend at northern mid-latitudes of about 1 DU per decade from Fig. 7, we are more confident in attributing high confidence to these detected negative trends.

When examining the global distribution of QR trends for the same datasets over the 2005-2021 time frame, we noticed overall larger differences between the datasets, with OMI-MLS showing mostly positive trends. This indicates that focusing on specific geographical regions and understanding the differences and the possible artefacts in the time series is required before we are able to provide a global picture of TrOC trends from satellite datasets.

**Table 2.** Trend values in ppb per decade in the five defined regions, over the period 2005-2021, for three datasets, with $2\sigma$ uncertainties, p-values, and respective trend confidence. Recommendations in Chang et al. (2023b) were followed.

| Region | Data set | Trend $\pm 2\sigma$ [ppb/dec] | p-value | confidence |
|---|---|---|---|---|
| West Africa | OMI-LIMB | $0.73 \pm 0.73$ | 0.31 | low |
| | SCIA+OMPS | $3.81 \pm 0.99$ | <0.01 | high |
| | OMI-MLS | $0.51 \pm 0.73$ | >0.33 | low |
| China | OMI-LIMB | $5.62 \pm 0.98$ | <0.01 | high |
| | SCIA+OMPS | $1.41 \pm 1.80$ | >0.33 | low |
| | OMI-MLS | $6.13 \pm 0.77$ | <0.01 | high |
| USA | OMI-LIMB | $-0.30 \pm 1.20$ | >0.33 | low |
| | SCIA+OMPS | $-3.26 \pm 1.59$ | 0.04 | medium |
| | OMI-MLS | $2.14 \pm 0.77$ | <0.01 | high |
| Mediterranean | OMI-LIMB | $-2.53 \pm 1.03$ | 0.01 | high |
| | SCIA+OMPS | $-1.42 \pm 1.93$ | >0.33 | low |
| | OMI-MLS | $0.60 \pm 0.99$ | >0.33 | low |
| Amazon | OMI-LIMB | $-0.43 \pm 0.90$ | >0.33 | low |
| | SCIA+OMPS | $4.84 \pm 1.46$ | <0.01 | high |
| | OMI-MLS | $4.41 \pm 0.99$ | <0.01 | high |

## 7 Conclusions

We performed an inter-comparison of several limb-nadir TOR datasets to assess their consistency, with a focus on finding approaches to look for similarities between them rather than highlight their differences.

The analysis revealed overall similarities in TrOC patterns across the datasets, such as the typical longitudinal asymmetry in the tropics and the maximum over Southeast Asia. However, notable differences in their seasonality, particularly in the southern hemisphere, have been highlighted. Some datasets show a drop in TrOC levels observed from 2020 onward, related by several studies to COVID-19 pandemic effects. This drop has been discussed by looking at possible discontinuities in the used TPH and reanalysis data without finding a convincing cause for it to be an artifact. Further investigations are required.

The biases due to the differences in TPH definition were found to be minor compared to other dataset-specific discrepancies; therefore, the effort to mitigate TPH-related biases in the level 3 data (gridded data) did not show a consistent reduction in the spread of TrOC values from different datasets. We suggest testing similar corrections in level 2 (daily or single profiles) data. We also found that trends in the TPH time series, used by the various satellite datasets to derive tropospheric ozone columns, are different from zero. These trends were also converted in terms of TrOC changes, finding the largest contribution

at mid-latitudes of up to +1 DU per decade, which needs to be kept in mind when computing TrOC trends.

Additionally, the drift in TOR products was assessed by collocating and comparing them to HEGIFTOM time series. We performed a latitude band-wise analysis, which revealed the presence of small but significant average drifts for some datasets within 0.6 DU per decade at mid-latitudes (excluding OMPS-MERRA) and up to 1.2 DU per decade in the tropics. These values need to be taken into account when evaluating TrOC trends, however, a direct subtraction of these drift values from TrOC trends is not straightforward. Satellite datasets in the tropics fail to capture short-term features shown by sonde data, which are most probably coming from the lowermost troposphere: a better characterization of the identified patterns is relevant for assessing the significance of TrOC trends. This analysis shows the value of satellite data providing global coverage and, at the same time, the need for stable long-term observations, e.g. ozonesondes, to assess drifts and discontinuities in their time series.

Our investigation of the TrOC trends over the 2005–2021 period focused on specific regions of interest rather than a global analysis. We applied a QR model including proxies to three long-period datasets and found consistent trends only in two areas: positive over China and negative over the Mediterranean, although with low confidence for the latter region. In most cases, the detected trends are well within $\pm 1$ DU per decade. The SCIA+OMPS dataset is affected in some regions by unexplained oscillations that make this product sub-optimal for trend studies.

Activities to homogenize these datasets in terms of TPH and a thorough comparison of satellite observations with ground-based data, consistently taking into account the different vertical resolutions, are recommended to better understand and reconcile the detected discrepancies between satellite datasets and interpret the trend results. Our analysis shows that a clear understanding of drifts and biases is crucial before using the datasets for global trends studies.

*Data availability.* GTO-LIMB and OMI-LIMB are available for download at https://nsdc.fmi.fi/data/data_sunlit.php (last access: 28.11.2024). OMI-MLS, OMPS-MERRA and EPIC-MERRA are publicly available at https://acd-ext.gsfc.nasa.gov/Data_services/cloud_slice/ (last access: 28.11.2024). The other two datasets are available upon request to the co-authors of this study. HEGIFTOM data are accessible through the following webpage: https://hegiftom.meteo.be/datasets/tropospheric-ozone-columns-trocs (last access: 28.11.2024).

**Appendix A**

Since the averaging kernels (AK) of TOC retrievals typically show a lower sensitivity to the lowermost troposphere, we tested the role of the close-to-surface ozone in terms of effects on the drift analysis (and trends, see Fig. B2). The application of satellite-specific AK is beyond the scope of this work; however, to take into account the different vertical sensitivity of the TOC measurements with respect to high-resolved ozonesondes, we tested a simple weighting of the sonde profiles before vertical integration, as described by a function providing a rough approximation of OMI averaging kernels:

$$w = \left( \frac{z}{z_{lim}} \right)^{0.5} \text{ for z} < z_{lim} \,, \tag{A1}$$

where z is the sonde altitude, $z_{lim}$ is two thirds of the TPH and the weight $w$ is equal to 1 above this altitude. Approximated OMI AKs are reported in the Supplements, Fig. S9. More examples of OMI AK can be found in e.g. Fig. S3 of Sofieva et al.

(2022). This paper also discusses the changes in TrOC retrieved by the residual method that are caused by the low sensitivity of nadir instruments in the lowermost troposphere.

Some caveats shall be added: the weighting from Eq. A1 does not conserve the TrOC value, so that this approach is suitable to investigate TrOC anomalies only. In addition, we are aware that the air mass factor in TOC retrieval takes into account the reduced sensitivity of the lowermost troposphere. This however might not be sufficient to capture cases with ozone concentrations in the boundary layer far from climatological values or cases when the TOC AK gets close to zero in the lowermost troposphere.

The relevant outcome of this weighting for the present manuscript is related to the drift of the satellite data in the tropics, shown in Fig. A1. In particular, by reducing the lowermost tropospheric contribution from the ozonesondes, drift values in the tropics are reduced (except for SCIA+OMPS). In addition, and most importantly, in comparison to panel (b) of Fig.8, satellite data better capture short-period patterns present in sonde data. In fact, the HEGIFTOM anomaly line (black) is less correlated to the TOR-HEGIFTOM residuals, indicating that reducing the weight of close-to-surface ozone improves the agreement with satellite data. This conclusion needs some further investigations.

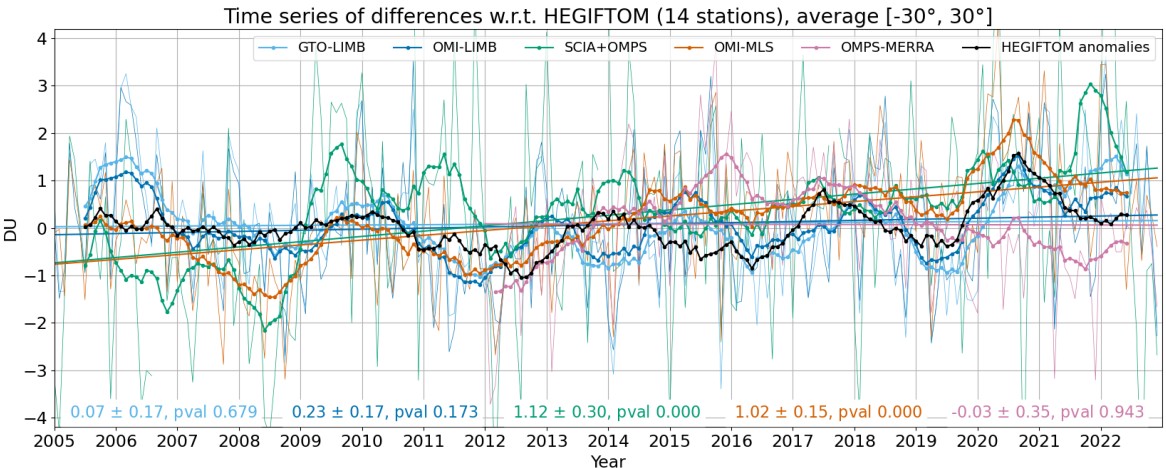

**Figure A1.** Drift in $\mathrm{DU/dec}$ of the difference time series between satellite and HEGIFTOM sonde anomalies in the tropics, similarly to Fig. 8 but including the weighting from Eq. A1. Thin lines indicate the monthly difference time series, whereas thicker lines are their 13-month averages; the HEGIFTOM anomalies are also shown (with sign changed). Linear trend values (drifts) with corresponding $2\sigma$ uncertainties and p-values are reported at the bottom of the panels. The titles include the number of stations used for the analysis.

## Appendix B

An overview of the comparison between ozonesonde data and the TOR products is given in Figs. B1 and B2, with the aim of providing an idea of the overall agreement between the datasets.

Figure B1 shows the relationship between the relative mean bias of each satellite dataset with respect to HEGIFTOM data (without AK application) on the y-axis and the respective standard deviation of the relative differences on the x-axis. Each dot

is a sonde station, and the color corresponds to its latitude. We notice the negative bias of the first two datasets with respect to most of the sonde TrOC values. On the contrary, S5P-BASCOE generally shows a high bias, particularly at northern mid-and high latitudes. The standard deviation of the differences tends to increase with latitude for most products, except for OMI-MLS, OMPS-MERRA and EPIC-MERRA. On the contrary, there is no indication of a dependence of the mean bias from latitude. We also notice the generally higher variability characterizing the SCIA+OMPS dataset.

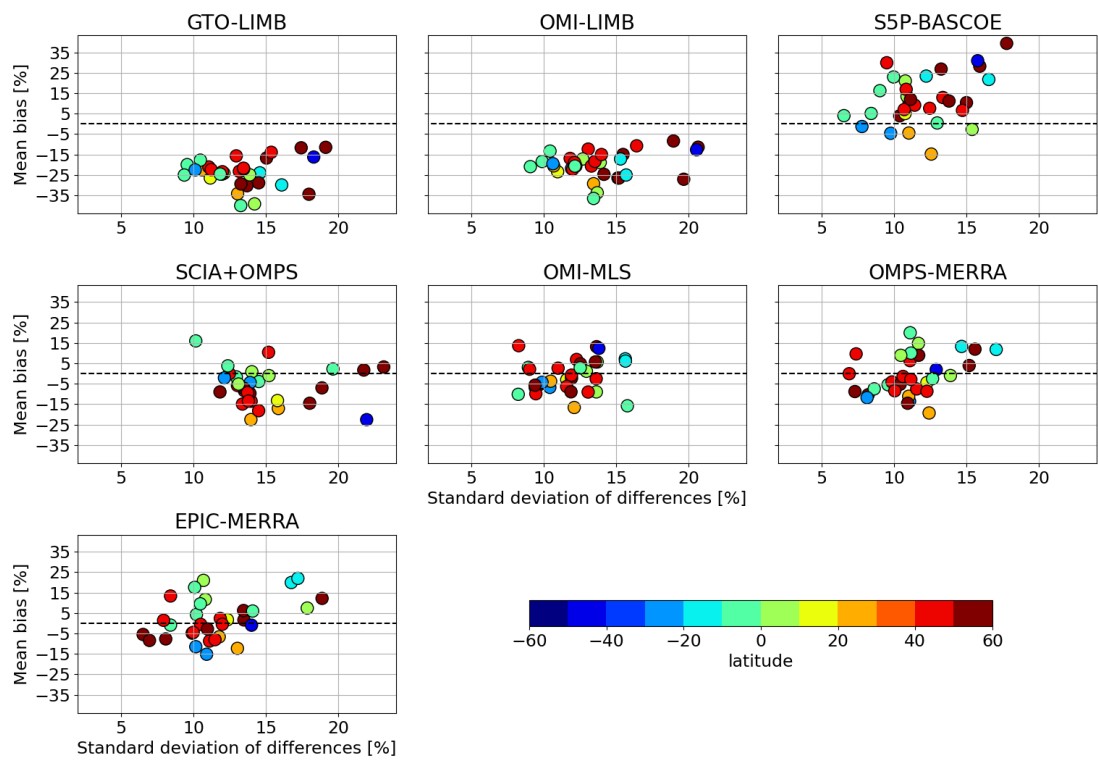

**Figure B1.** Scatter plot of the mean bias between HEGIFTOM data and each satellite dataset against the respective standard deviation of the differences, in percentage values. Each dot is a sonde station and the color corresponds to its latitude.

Figure B2 displays the trend values of the HEGIFTOM time series and of the collocated TOR products, in percentage per decade. These were computed by applying the QR model described in Sect. 6 to the relative anomalies of each time series. Only the three satellite products with the longest time series have been considered, taking into account that GTO-LIMB shows very similar results to OMI-LIMB so that it is not displayed here. For some stations, the trends do not show a good agreement between sondes and satellite data or even among the satellite products. We also include the HEGIFTOM trends computed

applying the AK weighting: these values generally show smaller discrepancies with respect to the satellite products and closer to zero trends overall.

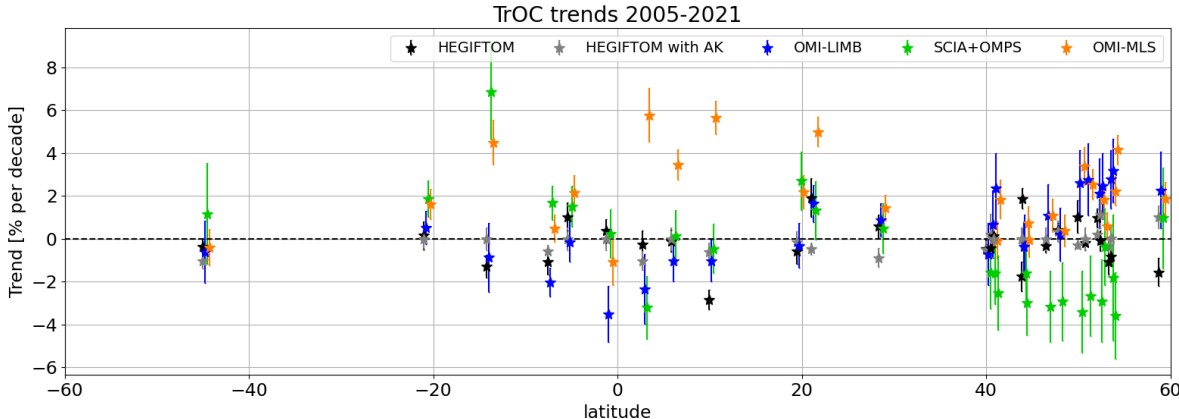

**Figure B2.** Trend values of the HEGIFTOM time series and of the collocated TOR datasets, in percentage per decade over 2005-2021. Only the three products with the longest time series are displayed (taking into account that GTO-LIMB results are very similar to OMI-LIMB)

*Author contributions.* CA performed most of the analysis for the inter-comparison of the datasets and wrote the manuscript. VS provided guidance and expertise for the analysis strategy and delivered the OMI-LIMB and GTO-LIMB datasets. AOC provided the SCIA+OMPS dataset and supported the work on the TPH correction. AR provided the expertise on limb-nadir matching and supervised the data analysis. KPH and DL delivered the S5P-BASCOE dataset. ED supervised the ESA-related OREGANO project. DT and RMS gave inputs and suggestions regarding the comparison with sondes. RVM gave support regarding the usage of the HEGIFTOM dataset. JZ delivered OMI-MLS, OMPS-MERRA and EPIC-MERRA and gave insights into the TrOC drop in recent years. MW, who leads the project, contributed to the scientific outcome. All co-authors contributed to the review of the manuscript.

*Competing interests.* Some co-authors are members of the editorial board of AMT.

*Acknowledgements.* This study has been primarily funded by ESA within the OREGANO (Ozone Recovery from Merged Observational Data and Model Analysis) project, and by the State and University of Bremen. GTO-LIMB and OMI-LIMB have been developed within the ESA CCI (Climate Change Initiative) project. We also acknowledge ESA for funding a six-month research stay for CA at the ESRIN Science Hub. Some of the ozonesonde data used in this publication are part of the Network for the Detection of Atmospheric Composition Change (NDACC) and are available through the NDACC website www.ndacc.org. We also acknowledge the data providers within SHADOZ and WOUDC networks. The processing of OMPS and SCIAMACHY stratospheric ozone and aerosol profiles has been done at the German NHR (National High Performance Computing) alliance, within the projects number hbk00063 and hbk00092. Part of the processing was also done on the Hypatia HPC facility at IUP, funded under DFG/FUGG grant nos. INST 144/379-1 and INST 144/493-1. The GALAHAD library was used for the processing. VS thanks the Academy of Finland (Centre of Excellence of Inverse Modelling and Imaging; decision no. 353082). We would like to thank Daan Hubert for the fruitful exchange of ideas and analysis approaches. We also acknowledge the contribution of the reviewers that helped improving the manuscript and the valuable advices from Owen Cooper.

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
