# Peer review of "Inter-comparison of tropospheric ozone column datasets from combined nadir and limb satellite observations"

_EGUsphere, 2024_

## Community Comment (CC1)

January 19, 2025

Comments by Owen R. Cooper (TOAR Scientific Coordinator of the Community Special Issue) on:

Inter-comparison of tropospheric ozone column datasets from combined nadir and limb satellite observations

Carlo Arosio, Viktoria Sofieva, Andrea Orfanoz-Cheuquelaf, Alexei Rozanov, Klaus-Peter Heue, Edward Malina, Ryan M. Stauffer, David Tarasick, Roeland Van Malderen, Jerry R. Ziemke, and Mark Weber

EGUsphere [preprint], https://doi.org/10.5194/egusphere-2024-3737
Discussion started Dec. 17, 2024
Discussion closes Jan. 22, 2025

This review is by Owen Cooper, TOAR Scientific Coordinator of the TOAR-II Community Special Issue. I, or a member of the TOAR-II Steering Committee, will post comments on all papers submitted to the TOAR-II Community Special Issue, which is an inter-journal special issue accommodating submissions to six Copernicus journals:  ACP (lead journal), AMT, GMD, ESSD, ASCMO and BG. The primary purpose of these reviews is to identify any discrepancies across the TOAR-II submissions, and to allow the author teams time to address the discrepancies.  Additional comments may be included with the reviews. While O. Cooper and members of the TOAR Steering Committee may post open comments on papers submitted to the TOAR-II Community Special Issue, they are not involved with the decision to accept or reject a paper for publication, which is entirely handled by the journal's editorial team.

**Comments regarding TOAR-II guidelines:**

TOAR-II has produced two guidance documents to help authors develop their manuscripts so that results can be consistently compared across the wide range of studies that will be written for the TOAR-II Community Special Issue.  Both guidance documents can be found on the TOAR-II webpage: https://igacproject.org/activities/TOAR/TOAR-II

*The TOAR-II Community Special Issue Guidelines*:   In the spirit of collaboration and to allow TOAR-II findings to be directly comparable across publications, the TOAR-II Steering Committee has issued this set of guidelines regarding style, units, plotting scales, regional and tropospheric column comparisons, and tropopause definitions.

*The TOAR-II Recommendations for Statistical Analyses*:  The aim of this guidance note is to provide recommendations on best statistical practices and to ensure consistent communication of statistical analysis and associated uncertainty across TOAR publications. The scope includes approaches for reporting trends, a discussion of strengths and weaknesses of commonly used techniques, and calibrated language for the communication of uncertainty. Table 3 of the TOAR-II statistical guidelines provides calibrated language for describing trends and uncertainty, similar to the approach of IPCC, which allows trends to be discussed without having to use the problematic expression, "statistically significant".

**General comments:**

In the introduction it would be helpful to cite some of the key papers from the first phase of TOAR as they are highly relevant to the background information on the importance of ozone for health, vegetation and climate: Fleming and Doherty et al. (2018), Mills et al. (2018), Gaudel et al. (2018).

Line 152
Regarding the impact of the COVID economic downturn on tropospheric ozone, there is now a large body of evidence that there was a widespread decrease of ozone in the free troposphere, and also at rural surface sites, of northern mid-latitudes. In addition to the papers already cited in this paper, you can also include: Miyazaki et al., 2021 (who show ozone decreases observed by the CrIS satellite instrument); Chang et al., 2022 show decreases above Europe; Putero et al. 2023, show ozone decreases at high elevation sites (a TOAR-II paper); see also Elshorbany et al., 2024 (a TOAR-II paper).

It would be helpful to compare your findings to those of other papers in the TOAR-II Community Special Issue. In particular, two papers discuss satellite-detected ozone trends: Pope et al., 2023 and Froidevaux et al., 2025.

Line 209
The authors make very good use of the HEGIFTOM database, which is a major achievement of the TOAR-II effort. However, there are two key time series in the HEGIFOTM database that were not used, and I recommend that they be included. As shown by Chang et al. (2024), recently published in the TOAR-II Community Special Issue, highly accurate quantification of free tropospheric ozone above a given location requires 15-20 profiles per month. There are only two locations in the world where ozone is profiled at such a high frequency. 1) Since 1994, the IAGOS commercial aircraft program has profiled the atmosphere above Frankfurt, Germany multiple times per day, and therefore this location has the most accurate monthly mean ozone profiles (see Figure 1, below). The data are so frequent that they can be matched to the overpass times of the satellite instruments. While the profiles only extend from the surface to 12 km, this covers the full depth of the troposphere for most of the year (winter, spring and autumn). 2) The JPL Table Mountain lidar (north of Los Angeles) has been sampling at the rate of 20 times per month since 2018. As the lidar has to operate under clear-sky conditions, it is an ideal match for UV satellite instruments. As shown in Figure 2, the lidar shows a drop in ozone in 2020, so it is an excellent time series for evaluating the impact of the COVID-19 economic downturn.

Following the TOAR-II Recommendations for Statistical Analyses, all trends need to be reported with their 95% confidence intervals and $p$-values. In the current draft, Figure 8, Figure 10 and Figure S10, Figure S12 and Figure S15 do not report $p$-values, and they need to be added.

**Specific comments**

Line 92
Here the period for the SCIA+OMPS product is listed as 2002-3023, but in Table 1 it is listed as 2004-2023. Why the discrepancy?

Line 109
Figure 1: It's worth pointing out that the only place where all products seem to agree well is northern mid-latitudes (40-60 N) in summer. Do you have an explanation? Why is there no comparison for the 20-40 N latitude band?

Line 120
It's not clear how the data sets were de-biased. Please explain.

Figure 4 and Figure S3
Please check the labels on the x-axis. OMI-MLS has been previously reported as showing a drop in ozone in 2020, but these figure show the drop occurred in 2021. Also, the time series in these plots seems to start in January 2006, when the OMI-MLS data go back to late 2004. Something seems to be off.

Line 251
A paper submitted to the TOAR-II Community Special Issue (Lu et al., 2024) quantifies ozone trends across East and Southeast Asia using ozonesondes and IAGOS data, and they found increases above both regions.

[Figure]

Figure 1. Ozone variability above Frankfurt, Germany, based on daily IAGOS commercial aircraft profiles, following the methods show in Figure 4 in Chang et al. (2022).

[Figure]

Figure 2. Ozone variability and trends based on the JPL Table Mountain lidar, following the methods show in Figure 3 in Chang et al. (2023). The mid-tropospheric 2000-2023 ozone trend is 0.98 ± 0.93 ppbv per decade (p=0.03).

**References**

Chang, K.-L., et al. (2022), Impact of the COVID-19 economic downturn on tropospheric ozone trends: an uncertainty weighted data synthesis for quantifying regional anomalies above western North America and Europe, AGU Advances, 3, e2021AV000542. https://doi.org/10.1029/2021AV000542

Chang, K.-L., Cooper, O. R., Rodriguez, G., Iraci, L. T., Yates, E. L., Johnson, M. S., Gaudel, A., Jaffe, D. A., Bernays, N., Clark, H., et al.: Diverging ozone trends above western North America: Boundary layer decreases versus free tropospheric increases, Journal of Geophysical Research: Atmospheres, 128, e2022JD038 090, 2023.

Chang, K.-L., Cooper, O. R., Gaudel, A., Petropavlovskikh, I., Effertz, P., Morris, G., and McDonald, B. C. (2024), Technical note: Challenges in detecting free tropospheric ozone trends in a sparsely sampled environment, Atmos. Chem. Phys., 24, 6197–6218, https://doi.org/10.5194/acp-24-6197-2024

Elshorbany, Y., Ziemke, J.R., Strode, S., Petetin, H., Miyazaki, K., De Smedt, I., Pickering, K., Seguel, R.J., Worden, H., Emmerichs, T. and Taraborrelli, D., 2024. Tropospheric ozone precursors: global and regional distributions, trends, and variability. ACP, 24(21), pp.12225-12257.

Fleming, Z. L., R. M. Doherty, et al. (2018), Tropospheric Ozone Assessment Report: Present-day ozone distribution and trends relevant to human health, Elem Sci Anth, 6(1):12, DOI: https://doi.org/10.1525/elementa.273

Froidevaux, L., Kinnison, D. E., Gaubert, B., Schwartz, M. J., Livesey, N. J., Read, W. G., Bardeen, C. G., Ziemke, J. R., and Fuller, R. A.: Tropical upper-tropospheric trends in ozone and carbon monoxide (2005–2020): observational and model results, Atmos. Chem. Phys., 25, 597–624, https://doi.org/10.5194/acp-25-597-2025, 2025

Gaudel, A., et al. (2018), Tropospheric Ozone Assessment Report:  Present-day distribution and trends of tropospheric ozone relevant to climate and global atmospheric chemistry model evaluation, Elem. Sci. Anth., 6(1):39, DOI: https://doi.org/10.1525/elementa.291

Lu, X., et al. (2024), Tropospheric ozone trends and attributions over East and Southeast Asia in 1995-2019: An integrated assessment using statistical methods, machine learning models, and multiple chemical transport models, submitted to ACP, EGUsphere [preprint], https://doi.org/10.5194/egusphere-2024-3702

Mills, G., et al. (2018), Tropospheric Ozone Assessment Report: Present-day tropospheric ozone distribution and trends relevant to vegetation, Elem. Sci. Anth., 6(1):47, DOI: https://doi.org/10.1525/elementa.302

Miyazaki, K., Bowman, K., Sekiya, T., Takigawa, M., Neu, J.L., Sudo, K., Osterman, G. and Eskes, H., 2021. Global tropospheric ozone responses to reduced NO x emissions linked to the COVID-19 worldwide lockdowns. *Science Advances*, *7*(24), p.eabf7460.

Pope, R. J., Kerridge, B. J., Siddans, R., Latter, B. G., Chipperfield, M. P., Feng, W., Pimlott, M. A., Dhomse, S. S., Retscher, C., and Rigby, R.: Investigation of spatial and temporal variability in lower tropospheric ozone from RAL Space UV–Vis satellite products, Atmos. Chem. Phys., 23, 14933–14947, https://doi.org/10.5194/acp-23-14933-2023, 2023.

Putero, D., et al. (2023) Fingerprints of the COVID-19 economic downturn and recovery on ozone anomalies at high-elevation sites in North America and western Europe, Atmos. Chem. Phys., 23, 15693–15709, https://doi.org/10.5194/acp-23-15693-2023

---

## Author Comment (AC1)

The study by Arosio et al. uses multiple satellite products of tropospheric column O3, derived from limb and nadir sounders, to investigate long-term trends in tropospheric ozone. Overall, this is a nice study and provides useful updates on tropospheric ozone trends, especially as part of TOAR-2. My main question is on Section 5 and Appendix A where the authors use a proxy of any OMI averaging kernel (AK) to investigate the impact of vertical sensitivity on comparisons to other datasets (i.e. HEGIFTOM) and long-term trends. I also have several minor comments listed below. Once these have been addressed, the manuscript is suitable for publication in AMT.

We thank the reviewer for the time spent on our manuscript and for his/her feedback. The comments are addressed below after each paragraph, and marked in blue; the suggested references were added to the paper. The manuscript has been accordingly modified. Line numbers in the answers refer to the updated manuscript.

**Major Comments:**

The authors attempt to investigate the impact of satellite averaging kernels on tropospheric column ozone trends by applying an approximation of an OMI AK to the ozonesonde data. So, firstly, the function in Equation 3 is based on what? Just the approximate shape of an OMI AK or a peer-reviewed study? Depending on the satellite product used (i.e. UV-Vis vs. IR or DOAS vs. optimal estimation), the shape of the AK profile can change substantially. So, would it not be worth trying to simulate the impact of multiple types of AKs? Also, to support your choice of Eqn 3, could you plot some actual OMI ozone Aks?

The reviewer addresses an important discussion topic and a weak point in the manuscript. Although interesting, the application of instrument-specific AK goes beyond the scope of this manuscript. We decided to restructure this part of the paper by moving the discussion related to the weighting of the ozonesonde profiles to resemble the sensitivity of nadir instruments to Appendix A. As a consequence, we left the drift assessment in Sect. 5 with the original TrOC time series provided by HEGIFTOM. In the Supplements we included a plot (corresponding to Fig. 1 below) of a typical AK from the OMI instrument, and added the Sofieva et al. (2022) reference. We slightly changed the former Eq. 3 (current Eq. A1) to better approximate the AK shape. This equation is a crude approximation of the typical OMI vertical sensitivity.

[Figure]

*Figure 1: OMI individual averaging kernels normalized to 1 above 25 km (grey lines) for clear-sky conditions on 1 July 2008, the mean averaging kernels (black lines) and their crude approximation (red dashed lines). Left: latitudes [1°S, 1°N], Right: latitudes [43°N, 47°N].*

**Minor Comments:**

Line 11: State which time-period the Southeast Asia trend is.

Sure, we stated that the trend is for the 2005-2021 period.

Line 17: Should the x in NOx be subscript?

Right, thanks.

Line 43-44: You state that ozone trends in Europe and North America has stabilised. This has been supported by several recent TOAR-2 studies (e.g. Pope et al., (2023 & 2024)), so might be worth mentioning them to support this statement.

Thanks, we included the suggested references in the introduction, line 46.

Line 50-52: Gaudel et al. (2018) identified large scale discrepancies between satellite product tropospheric ozone trends and you state that more work is needed to try and reconcile these. However, some TOAR-2 studies have attempted this, would be good to cite those (e.g. Gaudel et al., 2024 and Pope et al., 2024).

Thanks for the additional information, we included this in the manuscript, lines 55-56.

Figure 2: Could you add the global average trop O3 column average for each panel? While you can see the differences by eye, having a metric (e.g. average +/- standard deviation) above each map would add a useful overview for the reader.

We added this information only for the top panels, as the datasets in this plot have been debiased to the same multi-instrument mean. As a consequence, we provide in the figure a mean value with standard deviation for each season.

Equation 1: It is not clear what this is. Can this be defined?

We tried to clarify it by removing the subscript j and leaving only t, and modifying the explanation. The sentence reads now: "where Nm is the number of available monthly mean values TrOC(t) for each specific month of the year m, e.g. January, in each time series."

Line 136: What does m represent? It is not overly clear.

With m we indicate the generic month of the year, e.g. January.

Line 136-138: What are the offsets based on? Is this relative to the ozonesondes or the satellite ensemble average? Same for "after debiasing" in the Figure 3 caption.

Thanks, we specified in the text that the offsets are relative to the satellite average ensemble in both cases.

Line 144: "in spring time"…is this the boreal or austral spring?

We changed the text to "austral winter (JJA)".

Line 148: To make this clearer, could you add an example. E.g. (e.g. "where m indicates the month (e.g. January) of the year and tm all months m (e.g. all Januarys) in the time series").

Right, we included such an example in the description of Eqs. 1 and 2.

Figure 6e: What is causing the large standard deviation at approx. 20-30N?

We think the reviewer is here referring to the low standard deviation values found after the TPH correction in the northern sub-tropics. We verified that in this latitude band the TPH correction actually performs well, bringing together the mean values of the data sets. This indicates that at least in this region the biases between data are mostly caused by TPH discrepancies. We slightly changed the text at lines ~202 to take into account this comment.

Line 213: What do you mean by "after the harmonization of their time series by the HEGIFTOM"?

Right, this was a confusing statement and it was removed. We meant that the ozonesondes were harmonized by the HEGIFTOM working group.

Line 214: What data source is the thermal tropopause level based on?

We included in the text that the TPH was calculated, by the HEGIFTOM working group, using the temperature profiles from ozonesondes and the WMO thermal tropopause definition.

Equation 3: Add a reference, if appropriate, for this choice of function to represent the AK.

This was moved to Appendix A. We included in the Supplements (Fig. S9) a plot of the typical averaging kernels from OMI and the Sofieva et al. (2022) paper reference.

Line 243: Excess space between "last" and "10 years".

Sorry, we don't see that, probably a matter of spacing within the line.

Lines 244-246: I did not follow this text. This point probably needs more explanation.

The reviewer is right, we moved this section to Appendix A to better explain the point and expanded the discussion at line 257-258.

Lines 251-253: Would be useful to add the time-periods the authors derived trends for.

Thanks for the tip, we included the time frame of each study and expanded this paragraph.

Line 284-285: "The positive trend from SCIA-OMPS in the Amazon is likely related to artefacts in the datasets at the very beginning and end of the time periods". Can you expand on this. Not clear what you mean be artefacts and why this would drive an unrealistic trend.

We removed the sentence, but we meant that the larger anomalies towards the end of the SCIA+OMPS time series can influence the trend estimations.

Line 291: You discuss "insignificant trends at the 95% confidence level". I might be wrong, but I believe TOAR-2 are trying to move away from such definitions.

Thanks for this. It is true and also following Owen Cooper's comment we moved away from this formulation and introduced the p-values of the trends, together with their confidence level.

Line 293: If the time-series are influenced by "positive drift", since you can quantify this, could it not be removed to leave the real trends?

This is a good point, however we find such a subtraction not straightforward. First, a thorough drift investigation would be required to better target specific regions. This could provide a better characterization of the drift but is beyond the scope of this manuscript. In addition, the presence of

artifacts in the tropics in the drift analysis (see lines 257-258) gives less confidence in such a direct subtraction of the drift values for trends in specific regions.

Line 343: Why chose the OMI-LIMB product for this example?

This was chosen as it is one of the time series providing the longest record. OMI-MLS or GTO-LIMB could replace it without changing the conclusions of the paragraph.

**References:**

Gaudel, A., Bourgeois, I., Li, M., Chang, K.-L., Ziemke, J., Sauvage, B., Stauffer, R. M., Thompson, A. M., Kollonige, D. E., Smith, N., Hubert, D., Keppens, A., Cuesta, J., Heue, K.-P., Veefkind, P., Aikin, K., Peischl, J., Thompson, C. R., Ryerson, T. B., Frost, G. J., McDonald, B. C., and Cooper, O. R.: Tropical tropospheric ozone distribution and trends from in situ and satellite data, Atmos. Chem. Phys., 24, 9975–10000, https://doi.org/10.5194/acp-24-9975-2024, 2024.

Pope, R. J., Kerridge, B. J., Siddans, R., Latter, B. G., Chipperfield, M. P., Feng, W., Pimlott, M. A., Dhomse, S. S., Retscher, C., and Rigby, R.: Investigation of spatial and temporal variability in lower tropospheric ozone from RAL Space UV–Vis satellite products, Atmos. Chem. Phys., 23, 14933–14947, https://doi.org/10.5194/acp-23-14933-2023, 2023.

Pope, R. J., O'Connor, F. M., Dalvi, M., Kerridge, B. J., Siddans, R., Latter, B. G., Barret, B., Le Flochmoen, E., Boynard, A., Chipperfield, M. P., Feng, W., Pimlott, M. A., Dhomse, S. S., Retscher, C., Wespes, C., and Rigby, R.: Investigation of the impact of satellite vertical sensitivity on long-term retrieved lower-tropospheric ozone trends, Atmos. Chem. Phys., 24, 9177–9195, https://doi.org/10.5194/acp-24-9177-2024, 2024.

Sofieva, V. F., Hänninen, R., Sofiev, M., Szeląg, M., Lee, H. S., Tamminen, J., & Retscher, C. (2022). Synergy of using nadir and limb instruments for tropospheric ozone monitoring (SUNLIT). *Atmospheric Measurement Techniques*, *15*(10), 3193-3212.

---

## Author Comment (AC2)

General comments: This work provides an intercomparison of tropospheric ozone column datasets from combined nadir and limb satellite observations. Although this intercomparison is of interest to the community and deserves publication into the TOAR-II Special Issue, its presentation in terms of scientific clarity and focus could be substantially improved. Providing a more consistent story that is less broad would increase readability and significance. "The overall goal … to assess the consistency between the datasets and explore possible strategies to reconcile the differences between them" as phrased in the abstract does not seem to be fully (quantitatively) addressed, or the information is too scattered to be properly captured.

We thank the reviewer for the time spent on our manuscript and for his/her feedback. We reviewed the manuscript in light of this general comment, by making the overall structure more linear and consistent. For example, we reduced the content of the supplements, removed the analysis of global trends in Sect. 6 and focused the trend analysis only on quantile regression (QR).
Specific comments are addressed below after each paragraph, in blue text; the suggested references were added to the paper. The manuscript has been accordingly modified. Line numbers in the answers refer to the updated manuscript.

**Specific comments:**

- Line 22: "due to the overlap in signals" is too vague.
  We agree, this part of the sentence was deleted.
- Line 26: It would be appropriate to refer at least to ESA's operational TROPOMI nadir ozone product (possibly next to scientific products).
  Thanks, the reference to Keppens et al. (2024) was added.
- Line 56: Does "bias" here refer to bias from different TPH definitions or in general?
  It refers to the bias due to TPH definitions, it was clarified in the text: "a method to correct the tropopause definition-related bias between time series is presented and assessed." (line 60)
- Line 65: Does "its profile" refer to sensitivity or to stratospheric ozone?
  It refers to the stratospheric ozone profile, it was re-worded in the text: "as the ozone profile generally starts to increase below the typical thermal tropopause" (line 70)
- Lines 95-96: Was this drift correction done by the authors, or by the data providers? Please elaborate, possibly with reference(s).
  The correction was implemented by the data provider. This sentence was changed in the text: "To take into account an identified drift in OMI time series, this dataset was corrected by the data provider by adding a drift at a post-processing step, as described in Gaudel et al. (2024)."
- Line 98: "WMO thermal definition from NCEP reanalysis" requires explanation and references.
  We better defined the "WMO thermal definition" of TPH at this point and included a reference for it. (Line 103)
- Figure 1: Ordering the plots north to south would seem more logical, but is not mandatory. What happened to the 20-40 °N band plot?
  Thanks, it indeed makes more sense. We re-ordered the plots and slightly changed the latitude bands according to the TOAR II recommendations (the missing 20-40°N band is now included with the 20-30°N panel).
- Line 120 and following: Which global mean has been used as a reference for de-biasing?
  We used the multi-instrument mean. This was added in the text, line 128.

- Note sure whether the more qualitative Figure 2 adds a lot to 1 and 3?
  This is a good point, but we find interesting to show the TrOC patterns among the data sets and for different seasons to assess not only their latitude-averaged values but also their geographical ozone distributions.
- Line 164: Explain and explicitly indicate the "Quasi Biennial Oscillation signature"
  The reviewer is right, as it is not visible from this plot. We provide here a time series in the tropics, where the correlation of the satellite measurements with the QBO signal is remarkably high, as an example. This correlation was found to be higher for OMI-LIMB and GTO-LIMB than for OMI-MLS. This plot has been added in Fig. S7 (modified).

[Figure]

*Figure 1: GTO-LIMB time series in one lat-lon bin in the tropics, with its 13-month running average. The red line is the QBO contribution after fitting it using a MLR. The correlation coefficient is calculated between the fitted QBO and the 13-month average line.*

- Lines 168-169: "plays an important role in the biases between them" – How do you know, and doesn't this at least partially contradict conclusions made later, e.g. on line 178?
  Thanks for pointing this out, we replaced it with "it may play an important role..."
- Line 183: "As reference TPH, the ERA5 dataset was selected." This must refer to a different dataset than the monthly gridded ozone profiles mentioned in the previous sentence.
  It refers to the same ERA5 data set: in the previous sentence we describe the usage of ERA5 ozone profiles to compute the ozone column gaps due to the difference in TPH between the satellite dataset and ERA5, which is the reference value.
- Line 186: "We subtract the mean column gaps…" Is this the global mean, or determined for each latitude-longitude bin?
  This is determined for each latitude-longitude bin.
- Figure 7: More of interest than the trends themselves seem to be the differences in the trends due to different TPH definitions? The latter could be given more attention.
  This is an interesting point of discussion, we expanded the discussion in the manuscript, lines ~213-217. However, we think that the more relevant part is the TPH trend themselves, as they can have a direct impact on the TrOC trend from these datasets. The differences between the TPH are indeed related to different TPH definition, sampling of the data set or different reanalysis.
- Sections 5 and 6 refer to figures and tables that are distributed over the main text, appendices, and supplement, which hampers a fluent appreciation of the research and results by the reader.
  Thanks for pointing this out, we agree with the reviewer and implemented some changes in Sections 5 and 6 to reduce references to Appendix and Supplements (which were also reduced).

- Line 220: Possibly briefly explain the difference between drift and trend studies? It doesn't really help calling the drift plots in Figure 8 a "trend in DU per decade of the differences" and "trend values (drifts)"

  We introduce the concept of drift at line 238. The drift assessment involved computing a linear trend of the differences. In the caption of Fig. 8 we now say "Drift in DU/dec of the difference time series between satellite and HEGIFTOM sonde anomalies".

- Equation (3): Where does this come from? Is this a fully arbitrary choice, or is this based on common approaches in the literature?

  Thanks, we re-worked this section of the manuscript, moving this vertical weighting to Appendix A and added a reference (Sofieva et al. 2022) for the OMI AK.

- Line 231: Does "collocated" mean containing the station location?

  Yes, we replace "collocated" with "containing".

- Line 241: "with negative trends until around 2014 and positive trends in the last 10 years" Despite talking about drifts here (also see above), this is really not clear from Figure 8, especially as this figure only contains linear fitting to the full time series.

  Thanks, this has been changed, saying that the positive drift affecting most datasets in the tropics is larger during the last 10 years. However, the similarity between the SAT-HEGIFTOM (colored lines) and -HEGIFTOM (black lines) anomalies indicates that the patterns in sondes are not captured by the satellite datasets.

- Line 252: "trends of +(1-4)%/decade" over which period?

  We expanded this paragraph with a more detailed description of some studies at the beginning of Sect. 6.

- Figure 9: Do the regions defined here correspond to those agreed upon within the TOAR-II initiative?

  The TOAR II guidelines provide several options for the choice of regions, such as basic lat/long boxes around continents, HTAP regions, GBD regions and IPCC regions. We rather investigated the correlation of the seasonality between the data sets to define the specific regions, similarly to what was proposed by Van Malderen (2025).

- Line 268: Can you provide info and references on the multivariate regression model that is being used?

  We decided to remove the comparison between MLR and QR models and focused only on QR including proxies. This brought to a simplification of Sect. 6.

- Despite the anticipated focus on geographical regions in Section 6, Figure 11 extends to the global anyways. Possibly, limit this work to one of both only?

  The reviewer is right, we decided to remove this figure from the manuscript, to focus only on specific regions, and just point out in the manuscript the presence of larger differences when looking at the global picture.

- It would be very insightful to add an estimate (zonally) of the fraction of the trend that could be explained from the TPH trends in Figure 7?

  Thanks, in the trend discussion we introduced such a estimation, which helps the assessment of the confidence of the trend values.

- Line 298: "reconcile the discrepancies … rather than highlight the inherent differences" sounds conflicting. This requires some explanation of the distinction between both terms…

  With this sentence we meant to say that we didn't focus only on the discrepancies between the datasets, which have been presented for example in Figs.1 and 4, but rather on ways to

find similarities between the time series, in terms of climatological patterns (Figs. 2 and 3) and trends (Fig. 10). We modified this sentence by deleting "reconcile the discrepancies", to sound less contradicting.

- Line 319-320: "Our analysis shows that the homogenization is a crucial step before using the datasets for global trends studies." This is not what seems to come out of this work. Do you mean a global bias correction, a TPH correction, calculation of anomalies, or the ozonesonde homogenization for comparison? Both what this sentence is referring to as well as how this can be concluded from the analyses is very unclear, and makes this conclusion inappropriate. We agree that this sentence is confusing and sounds not well supported. We modified it as follows: "Our analysis shows that a better understanding of drifts and biases is a crucial step before using the datasets for global trends studies".

**Technical corrections:**

- Often inconsistencies occur between singular versus plural nouns and verbs. Please verify throughout the text.
  Thanks, we performed a review of the English.
- Line 9: "morphology" does not seem to be the right term here, or should be explained.
  We replaced "morphology" with "distribution patterns".
- Line 91: Rephrase "a DU dataset"
  "a DU dataset" → "a dataset in DU".
- Line 105: Add monthly temporal resolution?
  We added this information at line 110.
- Lines 108-117: "overall" is used quite often, sometimes inappropriately, i.e., conflicting with other parts of the sentence(s), e.g. "zonal averages" in line 108.
  Thanks, we reviewed the usage of "overall".
- Caption of Figure 4: "De-seasonalized anomaly time series…" ?
  We removed "anomaly".
- The color scale of Figure 6 (b) is hardly visible in print.
  Thanks, the scale has been improved.

Additional references

Arosio, C., Rozanov, A., Malinina, E., Weber, M., & Burrows, J. P. (2019). Merging of ozone profiles from SCIAMACHY, OMPS and SAGE II observations to study stratospheric ozone changes. *Atmospheric Measurement Techniques*, *12*(4), 2423-2444.

Keppens, A., Di Pede, S., Hubert, D., Lambert, J. C., Veefkind, P., Sneep, M., ... & Zehner, C. (2024). 5 years of Sentinel-5P TROPOMI operational ozone profiling and geophysical validation using ozonesonde and lidar ground-based networks. *Atmospheric Measurement Techniques*, *17*(13), 3969-3993.

Sofieva, V. F., Hänninen, R., Sofiev, M., Szeląg, M., Lee, H. S., Tamminen, J., & Retscher, C. (2022). Synergy of using nadir and limb instruments for tropospheric ozone monitoring (SUNLIT). *Atmospheric Measurement Techniques*, *15*(10), 3193-3212.

Van Malderen, R., Thompson, A. M., Kollonige, D. E., Stauffer, R. M., Smit, H. G., Barras, E. M., ... & Sussmann, R. (2025). Global Ground-based Tropospheric Ozone Measurements: Reference Data and Individual Site Trends (2000–2022) from the TOAR-II/HEGIFTOM Project. *Atmospheric Chemistry and Physics*.

---

## Author Comment (AC3)

We thank Owen Cooper for the time he spent reviewing the manuscript and for the constructive comments. We revised the manuscript in light of the TOAR II Guidelines, in terms of color codes, statistical practices and units. We address his specific comments below (blue text).

**General comments:**
In the introduction it would be helpful to cite some of the key papers from the first phase of TOAR as they are highly relevant to the background information on the importance of ozone for health, vegetation and climate: Fleming and Doherty et al. (2018), Mills et al. (2018), Gaudel et al. (2018). Thanks for pointing this out, Fleming et al. (2018) and Mills et al. (2018) have been added as well in the introduction.

Line 152
Regarding the impact of the COVID economic downturn on tropospheric ozone, there is now a large body of evidence that there was a widespread decrease of ozone in the free troposphere, and also at rural surface sites, of northern mid-latitudes. In addition to the papers already cited in this paper, you can also include: Miyazaki et al., 2021 (who show ozone decreases observed by the CrIS satellite instrument); Chang et al., 2022 show decreases above Europe; Putero et al. 2023, show ozone decreases at high elevation sites (a TOAR-II paper); see also Elshorbany et al., 2024 (a TOAR-II paper).
Thanks for these references, we expanded the paragraph to include these works.

It would be helpful to compare your findings to those of other papers in the TOAR-II Community Special Issue. In particular, two papers discuss satellite-detected ozone trends: Pope et al., 2023 and Froidevaux et al., 2025.
If the comment is related to Sect. 6 and TrOC trends, we shortly summarized the suggested findings there.

Line 209
The authors make very good use of the HEGIFTOM database, which is a major achievement of the TOAR-II effort. However, there are two key time series in the HEGIFOTM database that were not used, and I recommend that they be included. As shown by Chang et al. (2024), recently published in the TOAR-II Community Special Issue, highly accurate quantification of free tropospheric ozone above a given location requires 15-20 profiles per month. There are only two locations in the world where ozone is profiled at such a high frequency. 1) Since 1994, the IAGOS commercial aircraft program has profiled the atmosphere above Frankfurt, Germany multiple times per day, and therefore this location has the most accurate monthly mean ozone profiles (see Figure 1, below). The data are so frequent that they can be matched to the overpass times of the satellite instruments. While the profiles only extend from the surface to 12 km, this covers the full depth of the troposphere for most of the year (winter, spring and autumn). 2) The JPL Table Mountain lidar (north of Los Angeles) has been sampling at the rate of 20 times per month since 2018. As the lidar has to operate under clear-sky conditions, it is an ideal match for UV satellite instruments. As shown in Figure 2, the lidar shows a drop in ozone in 2020, so it is an excellent time series for evaluating the impact of the COVID-19 economic downturn.
Thank you for suggesting these ideas. However, we did not implement them in the present work for the reasons explained below. First, the focus of this manuscript is to inter-compare satellite datasets, not to perform a validation of them: the ground-based observations are for this work a reference for the comparison, especially in terms of long-term drift. Secondly, using the Table Mountain Lidar would be useful for our manuscript only to further investigate the COVID-related drop. We did not expand further on this topic as it is not the main focus of the paper, we just wanted to mention how the different data sets show this drop. In this regard, one of the reviewers suggested the need to narrow the focus of the investigation without branching out too much in side topics and Supplementary information. Third, we had a look at the IAGOS data, for example for Frankfurt, and it is indeed an

interesting high sampled data set, but it rarely reaches up to the thermal tropopause height, also during winter months. A direct comparison of the IAGOS data set with the monthly satellite data would require some adjustment of the used satellite datasets with the help of ERA5 (or other reanalysis data) to bridge the gap between the monthly averaged TPH and IAGOS top altitude. This procedure goes also beyond the scope of this manuscript and we think it would introduce more discussion on the proper methodology (i.e., the quality of this adjustment) rather than provide a better information on the quality of the satellite data.

Following the TOAR-II Recommendations for Statistical Analyses, all trends need to be reported with their 95% confidence intervals and p-values. In the current draft, Figure 8, Figure 10 and Figure S10, Figure S12 and Figure S15 do not report p-values, and they need to be added.
Thanks for this point, we removed the "statistical significant" expression, adapting the terminology as recommended in the Guidelines, and included the p-values in the pictures and Table 2.

**Specific comments**
Line 92
Here the period for the SCIA+OMPS product is listed as 2002-3023, but in Table 1 it is listed as 2004-2023. Why the discrepancy?
Thanks, the discrepancy was corrected.

Line 109
Figure 1: It's worth pointing out that the only place where all products seem to agree well is northern mid-latitudes (40-60 N) in summer. Do you have an explanation? Why is there no comparison for the 20-40 N latitude band?
We changed the plot to adapt the latitude bands to the TOAR II recommendations. We mention the agreement at northern mid-latitudes in the description of Fig.1.

Line 120
It's not clear how the data sets were de-biased. Please explain.
Thanks, this is now better explained in the manuscript. The de-biasing simply consists in the removal of the offset between the specific satellite product and the multi-instrumental mean.

Figure 4 and Figure S3
Please check the labels on the x-axis. OMI-MLS has been previously reported as showing a drop in ozone in 2020, but these figures show the drop occurred in 2021. Also, the time series in these plots seems to start in January 2006, when the OMI-MLS data go back to late 2004. Something seems to be off.
Thank you, we have checked and the time lines are consistent. The drop is not particularly visible in OMI-MLS in 2020 because the anomalies are computed over the entire time period. As shown in Fig. S3 when the anomalies are computed over the 2016-2019 period, the drop becomes more evident also in 2020 and in other data sets. The plot was updated with start date in 2004 instead of 2006.

Line 251
A paper submitted to the TOAR-II Community Special Issue (Lu et al., 2024) quantifies ozone trends across East and Southeast Asia using ozonesondes and IAGOS data, and they found increases above both regions.
Thanks, we included this in Sect. 6 and in the references.

---

## Author Response (AR2)

Thanks to the Editor for the positive feedback. We addressed the technical corrections as following addressed (in blue).

- Line 19: This reads as if trop. O3 is just caused by anthropogenic emissions, which is not the case. Natural sources of O3 precursors need to be added.
  We included the suggestion as follows: "The primary sources of tropospheric ozone precursors are both natural, e.g. wetland methane emissions, wildfires and lightning, and anthropogenic, e.g. vehicle emissions, industrial activities, and chemical solvents, making ozone a critical focus for air quality regulations and environmental health initiatives (Brown et al., 2013)."

- Line 28: Propose to reword to "and the limited information content of nadir observations for vertical profiles."
  We re-worded the sentence as follows: "However, distinguishing between stratospheric and tropospheric ozone is challenging due to the low amount of ozone in the troposphere in comparison to the stratosphere, and the limited information content of nadir observations for vertical profiles."

- Line 70: "Satellite datasets are in this study" -> "In this study, satellite datasets are …"
  Addressed as suggested.

- Line 173: "ruling out" is a too strong statement. Change to "no indications of TPH-related artifacts could be observed" or sth similar.
  We re-worded the sentence as follows: "Also in TPH anomalies no evident discontinuities have been found around 2020 (as shown Fig.~S4), without indications of TPH-related artifacts."

- Fig. A1: The caption should relate this to Fig. 8 and point out what is different here.
  Right, we included this point in the caption: "Drift in DU/dec of the difference time series between satellite and HEGIFTOM sonde anomalies in the tropics, similarly to Fig. 8 but including the weighting from Eq. A1."